# Physical Reasoning and Object Planning for Household Embodied Agents

**Ayush Agrawal**                                    *ay.agrawal812@gmail.com*
*National University of Singapore*

**Raghav Prabhakar**                                 *raghav.prabhakar66@gmail.com*
*IIIT-Hyderabad, India*

**Anirudh Goyal**                                    *anirudhgoyal9119@gmail.com*
*Deepmind, London*

**Dianbo Liu**                                       *dianbo@nus.edu.sg*
*National University of Singapore*

**Reviewed on OpenReview:** `https://openreview.net/forum?id=xYkdmEGhIM`

## Abstract

In this study, we explore the sophisticated domain of task planning for robust household embodied agents, with a particular emphasis on the intricate task of selecting substitute objects. We introduce the **C**ommonSense **O**bject **A**ffordance **T**ask **(COAT)**, a novel framework designed to analyze reasoning capabilities in commonsense scenarios. This approach is centered on understanding how these agents can effectively identify and utilize alternative objects when executing household tasks, thereby offering insights into the complexities of practical decision-making in real-world environments. Drawing inspiration from factors affecting human decision-making, we explore how large language models tackle this challenge through four meticulously crafted commonsense question-and-answer datasets featuring refined rules and human annotations. Our evaluation of state-of-the-art language models on these datasets sheds light on three pivotal considerations: 1) aligning an object's inherent utility with the task at hand, 2) navigating contextual dependencies (societal norms, safety, appropriateness, and efficiency), and 3) accounting for the current physical state of the object. To maintain accessibility, we introduce five abstract variables reflecting an object's physical condition, modulated by human insights, to simulate diverse household scenarios. Our contributions include insightful human preference mappings for all three factors and four extensive QA datasets (2K, 15k, 60k, 70K questions) probing the intricacies of utility dependencies, contextual dependencies and object physical states. The datasets, along with our findings, are accessible at: `https://github.com/com-phy-affordance/COAT`. This research not only advances our understanding of physical commonsense reasoning in language models but also paves the way for future improvements in household agent intelligence.

# 1 Introduction

Humans, as beings innately attuned to their surroundings, traverse a world where conversations, decisions, behaviors, and understanding are deeply embedded in the underlying fabric of a situation. Their engagement with the world entails commonsense (background) knowledge about entities–properties, spatial relations, events, causes and effects, and other social norms ((McCarthy, 1959); (Winograd, 1972); (Davis & Marcus, 2015)). The importance of situational awareness is starkly evident in our daily tasks, where choosing objects for specific activities showcases our adaptability to different settings. Consider the straightforward task of cutting a cake—how do we determine which object is suitable for this task? When a person needs to select an object to accomplish this task, an array of factors will affect the choice. For example, we must choose something capable of cutting (*Utility*)[1], suitable for cutting a cake (*contextual appropriateness*), and likely in an appropriate physical condition to be used (*physical state*). These considerations would be to ensure the appropriateness, ease, and safety of those cutting the cake as well as who will eat the cake. Although these considerations might seem trivial and intuitive to us humans, they are still an important aspect to consider when developing embodied household agents. Such reasoning capabilities can be potentially leveraged by embodied agents to generate action plans for human requirements represented in natural language. In this work, we propose a **C**ommonSense **O**bject **A**ffordance Task: a textual physical commonsense task to evaluate most appropriate object selection capabilities in the presence of various alternative objects.

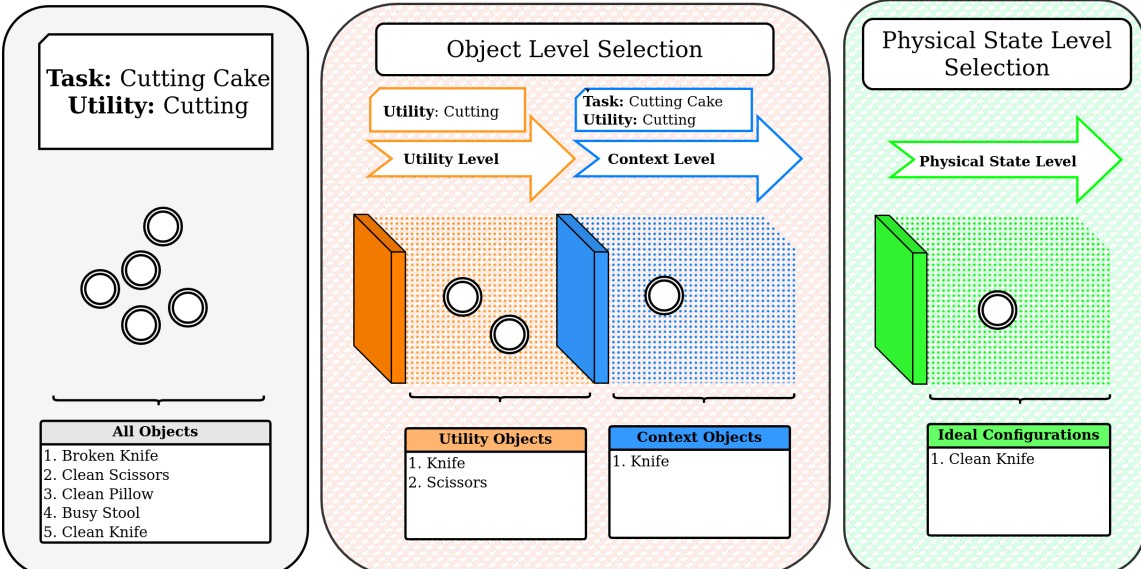

Figure 1: We divide the whole decision-making process into 2 broad phases. Pruning out options firstly based on **Object Level** then **Physical State**. Within the Object level, we further divide into 2 sub-steps: **Utility** and **Contextual Appropriateness**. We highlight this method's adeptness in comparing appropriateness across an array of factors and coming up with a substitute object even in the absence of the ideal object [*Cake Knife*]. Our work provides QA datasets about this type of commonsense reasoning

Recent advancements in large language models (LLMs) (Zhu et al., 2023; Peng et al., 2023; Zhang et al., 2023; Brown et al., 2020; Chowdhery et al., 2022; Touvron et al., 2023; OpenAI, 2023) have significantly enhanced our ability to extract rich commonsense knowledge from extensive web data. To analyze this task and evaluate the current capabilities of Language Models across such human commonsense-oriented reasoning, we develop this task as a decision-making process spanning 3 major aspects: **Utility**: The concept of *Utility*, a focal point in previous research Speer et al. (2017), elucidates our understanding of an object's functionality in a variety of situations. Although ConceptNet (Speer et al., 2017) has been a crucial tool for identifying object-utility relationships, its nature as a human-compiled knowledge graph has led to the pursuit of more dynamic sources. We curate Object-Utility mappings pertaining to this aspect and a 2K QA dataset to evaluate the object selection capabilities of language models based on required utility. **Context**: Our decision-making extends beyond mere utility. To account for the various situational factors such as safety, social norms adherence, effort optimization, efficiency, and situational appropriateness, we introduce the second aspect as: *contextual appropriateness* This adeptness in judgment arises from our ingrained commonsense, sculpted by experience and intuitive physical understanding. To evaluate the reasoning capabilities of various language models across this aspect, we generate Object-Utility-Task mappings and curate a 15K MCQ QA dataset. **Physical State**: Previous work Li et al. (2023) has shown how object choice depends on various physical variables. To make this aspect more human commonsense-oriented, we add a layer of abstraction and introduce 5 abstract variables to depict the current physical state of the object. To observe how object usability evolves with various abstract physical state variations, we generate human preference mappings and curate 2 QA datasets (specifically focused on analyzing object usability with varying physical states). These two physical state datasets summed up to 130K MCQ question-answer pairs combined. Thus overall, we curated 4 QA datasets.

In Figure 1, we illustrate an example of using these 3 aspects to select the best feasible object. For the task of cutting a cake where the following objects are available: a Broken Knife, Clean Scissors, a Clean Pillow, and a Clean Knife. Beginning with pruning objects based on their utility(cutting) brings our focus primarily on Knife and Scissors. Further knowledge about the task (cutting a cake) leads to the dismissal of Scissors as a suitable tool for cake cutting. Finally, upon considering the physical state of Knives, the Clean Knife emerges as the obvious choice. This explains the three key factors we explore for evaluating task-specific object selection capabilities – the ***utility of the object***, its ***contextual appropriateness***, and its ***current physical state***.

Such commonsense reasoning capabilities not only allow us to judge the appropriateness of an object in the context of the given task but also help us in successfully coming up with an appropriate substitute object in the absence of the most ideal object(Here: *Cake Knife*). Such skills, if equipped with embodied agents, will enhance their reasoning capabilities and make them adept in planning tasks in scenarios where the ideal object is not available. .

**Main Contributions**: In this study, we made the following contributions:

- Creation and provision of human-preference mappings across all 3 aspects of the **C**ommonSense **O**bject **A**ffordance **T**ask(COAT)

---

[1]This shouldn't be confused with the overall objective of choosing an object that maximizes the utility. This could be comprehended as "function" or "aspect" in the focus of the given task

- Introduction of 4 major novel CommonSense-based QA Datasets, facilitating an in-depth analysis of how object usability evolves under different utility requirements, contextual scenarios, and physical states
- Evaluation of Large Language Model baselines on these datasets, accompanied by a detailed analysis of their performance in multi-step abstract reasoning scenarios.

## 2 Dataset Creation

To systematically investigate the capacity of LLM to conduct human-style physical commonsense reasoning and preferences across three crucial factors, we have devised an experimental framework centered around 75 household tasks, carefully curated to span 22 distinct utilities. The experiment involves a diverse inventory of 100 objects sourced from the AI2Thor Simulator (Speer et al., 2017), ensuring relevance and diversity within a household context.

1. `Tasks`: are high-level household activities that could be accomplished by a human or embodied agent. Example: `Cutting a Cake`. See Task List

2. `Utilities`: are different aspects of a high-level task. A task can comprise 1 or more utilities. For the example of `Cutting Cake`, the utility could be `Cutting`. While for the task of `Making an Omelette`, utilities could be `Mixing`, `Heating` etc. See Table 2a

3. `Objects`: are a subset of objects available in AI2Thor (Kolve et al., 2022) Simulator. See Table 2b

| *Utilities* | | | *Objects* | | |
|---|---|---|---|---|---|
| Carrying | Comfort | Heating(vessel) | Bowl | Bed | Pan |
| Cleaning | Washing | Mixing(tool) | DishSponge | SinkBasin | Spoon |
| Disposing | Cutting | Mixing (vessel) | GarbageCan | Knife | Cup |
| Storage | Entertainment | Heating(source) | Fridge | Laptop | Microwave |
| Reading | Breaking | Increasing Height | Newspaper | BaseballBat | Chair |
| Eating | Writing | Physical Activity | Apple | Pen | Dumbell |
| Decoration | Light Source | Surface Support | Statue | Floor Lamp | CounterTop |

(a) A representational subset of utilized Utilities  (b) A representational subset of utilized Objects

The following section gives an overview of the annotation tasks and the process of creating CommonSense Reasoning Datasets.

### 2.1 Human Preference Collection

### 2.1.1 Utility

Incorporating GPT3.5-turbo (Brown et al., 2020) along with human commonsense annotations, we meticulously established a mapping between utilities and objects. These are called `Utility Objects`. Notably, each object may be associated with multiple utilities, and conversely, a single utility can

be linked to various objects. Table 8 provides an overview of the utilities along with their associated objects utilized in our experiments. More Information about the annotation process can be found in Appendix D

### 2.1.2 Contextual Appropriateness

In evaluating object utility, it is crucial to recognize that suitability for specific tasks can vary significantly. Take, for example, the multifaceted use of a candle. While it possesses the inherent ability to generate heat, employing a candle for heating soup introduces a range of practical limitations. This observation underscores the complexity of human preference and decision-making in the context of object utility. Key factors influencing these choices include efficiency (as illustrated by the impracticality of using a candle for heating soup), safety considerations (such as the risks associated with standing on an armchair), social norms and constructs (exemplified by the unconventional choice of serving wine in a bowl), and the overall appropriateness of an action (e.g., the disposal of eggshells in a sink basin). To systematically explore these dynamics, we engaged human annotators in a study designed to assess the selection of appropriate objects for specified tasks and utilities

### 2.1.3 Physical State

The selection of objects for specific tasks is influenced not only by intangible factors such as safety and social constructs but also by the object's current physical state. Prior research, including the works of Li et al. (2023) and Gao et al. (2023), has employed various physical parameters to examine Large Language Models' (LLMs) comprehension of an object's physical attributes. In our study, we shift the focus to task planning under non-ideal conditions, necessitating reasoning about potential substitute objects. To this end, we have developed five distinct variables, each represented by abstract symbolic terms. These variables have been derived directly from the AI2Thor Simulator, facilitating their broader applicability and potential integration into the burgeoning field of Embodied AI. Table 1 delineates these variables and their corresponding abstract values. Here, `Already In Use` variable is used to represent the availability of an object for use. Some examples of an object in `reversible-using` state are the object getting recharged, a wet object, or an object in a reversible state (meaning it will need time to get back to the ideal state or is temporarily being used by someone else). Whereas in an `irreversible-using` state, the object could be broken, depleted, out of stock, and thus is in an irreversible state of use. Further details about the chosen physical variables are elaborated in Appendix A.1

| Variables | Abstract Values |
|---|---|
| *material* | Metal, Wood, Plastic, Glass, Ceramic, Stone, Wax, Fabric, Rubber, Food, Paper Sponge, Organic, Soap |
| *mass* | Light, Medium, Heavy, Super-Heavy |
| *temperature* | Cold, RoomTemp, Hot |
| *already in use* | Free, Reversible-Using , Irreversible-Using |
| *condition* | Dirty, Clean, Broken |

Table 1: Abstract Values for Various Variables

**Gathering Common Object Configurations** In the context of this study, a `Configuration` denotes the physical state of an object characterized by five variables. While a wax chair might be conceivable in the realm of Madame Tussauds, it remains highly improbable in everyday household scenarios. Thus, to ensure the relevance of configurations to common household scenes, human annotators were tasked with selecting plausible and frequently occurring variable values for each object. (See Appendix D)

**Ranking Object Configurations** In our study, we not only provided configurations that occur commonly but also tasked the annotators with categorizing the configurations of an object into three distinct classes: `Ideal`, `Moderate`, and `Bad`. This classification was predicated on their assessment of the anticipated time required for an agent to commence the task with a given object configuration. Utilizing these categorizations, we constructed two comprehensive datasets comprising 130,000 questions specifically designed to assess the physical commonsense reasoning capabilities of Large Language Models. Further details on this process are elaborated in Appendix D

## 2.2 CommonSense QnA Datasets

Based on `Utility Appropriateness`, `Contextual Appropriateness`, and `Physical State`. We created 4 CommonSense QA datasets.

1. **Task-u**[2]: This experiment was based on pruning objects based on their compatibility with the utility specified in the question. We curated a **Object$_{utility}$ Level QA Dataset** and utilized **Object-Utility Mappings** (obtained from human annotators) for setting the ground truth.

2. **Task-0**: This experiment was based on pruning objects based **only** on contextual factors that affect an object's task-specific appropriateness. We curated another **Object$_{context}$ Level QA Dataset** and used **Context Mappings** for setting the ground truth.

3. **Task-1** & **Task-2**: These experiments were based on pruning out objects based on **only** physical state variations (described by 5 symbolic variables). We curated 2 **Variable Level Datasets** and utilized human annotations for setting the ground truth.

### 2.2.1 Object$_{utility}$ Level Dataset

To evaluate the object-utility alignment in LLMs, we curated an Object Level QA dataset (with ground truth obtained from human annotations) and made LLMs choose the most appropriate object for a given utility. Here we specified no information about the context and physical state of objects. This was done to evaluate solely utility-based selection capabilities.

---

**Example Question of Task-u**

**Question**: Which of the following objects would be best suited for the purpose of "heating(source)"?
**Options**:
    (A) Spatula
    (B) StoveBurner
**Correct Answer**: B

---

[2]These are different from the tasks(activities) used to curate datasets

### 2.2.2 Objectcontext Level Dataset

To evaluate the reasoning capabilities of LLM when choosing objects over contextual factors, we curate an Object Level QA dataset. Here, previously recorded `Context Mappings` were kept as ground truth. (See Annotation Task 2.1.2). Here, we specified no information about the physical state, thus assuming every object to be an ideal configuration. This was done to create QnA datasets focused solely on object selection capabilities based on contextual factors.

**Question**  Every question can be assigned a <`Task`, `Utility`> combination and was framed in the way shown below:

---

**Question**

What object would you be choosing for <**utility**> when you are tasked to <**task**>?

---

**Options**  Based on the sampling strategy and the number of options in the prompt, we created **4** variations of objectcontext `level dataset`. An example of such variation is shown below.

1. **Variation-1** : For each question, we randomly sampled **1 context object** and **1 utility object** both belonging to the same `utility`.[3]

---

**Example Task 0 Variation 1**

**Question ID**: 1,
**Utility**: heating(source),
**Question**: Which of the following objects would be best suited for the purpose of "heating(source)" when tasked to "reheating coffee"?
**Options**:
   (A) Toaster
   (B) StoveBurner
**Correct Answer**: B

---

### 2.2.3 Physical Configuration Level Dataset

Based on `Common Configurations` generated in the annotation task [2.1.3], we create 2 Variable Level QA datasets to analyze the reasoning capabilities of Language Models on pruning out options based on their current physical state. The 2 datasets differ in the level of difficulty and the level of reasoning required to answer the questions correctly. We describe the creation process in this section. The question in both datasets remains the same as that of Object Level Dataset. However, unlike the first dataset where the options were objects, this time we give various `Configurations` of `Context Objects` as options. Here we ensured all options were appropriate according to the question's utility and context. This was done to evaluate the object selection capabilities solely based on the physical state, thus precluding the possibility of wrong answers due to a wrong object being selected (due to the other 2 factors).

---

[3]Details about other variations for Task 0,1,2 can be found here Appendix B

---
**Configuration**

**object name**: Microwave, **mass**: super-heavy, **temperature**: RoomTemp, **material**: Metal **already in use**: free **condition**: clean

---

For $\text{Object}_{\text{utility}}$ `level Dataset`, we sampled <Utility Objects> based on the question's utility and for $\text{Object}_{\text{context}}$ `level Dataset`, we sampled <Context Objects> based on question's <Task,Utility> combination. However, we use a different approach for generating physical configuration datasets. Here, we classified the configurations of Context Objects into three broad categories: "Ideal," "Moderate," and "Bad." Each category is defined by specific variable values that delineate their characteristics. The "Ideal" category represents configurations in their optimal states, facilitating the specified task without additional time/material penalties. In contrast, the "Moderate" category includes configurations that deviate from these ideal states, resulting in both time and material costs for their utilization. The models assess these options based on their estimated costs. Lastly, the "Bad" category comprises configurations that make the Context Objects unusable (even after considering potential penalties). Both "Moderate" and "Bad" configurations are grouped under Sub-Optimal Configurations, offering a nuanced understanding of the varying degrees of object usability.

By sampling options from these 3 sets of configurations [2.1.3], we divide our efforts into creating 2 physical configuration datasets:

**A. Ideal Configuration Dataset**  In alignment with its name, the "Ideal Configuration" dataset involves questions with the correct answer as `Ideal Configuration` of `Context Object` of the question's associated <Task,Utility> combination. To systematically analyze the behavior of models, we introduce 12 distinct variations of this dataset. The creation of these variations is designed to progressively augment the complexity of the datasets, facilitating a comprehensive analysis of model behaviors. Each of the 12 variations comprises approximately 5,000 question-answer pairs, with differing counts of options—ranging from 5 options to 2 options per question. Along with the varying number of options, we also ablated on various sampling techniques. While different sampling techniques help us study their behavior concerning different object distributions, the deliberate variation in the number of options enables us to evaluate the success rate variations of Large Language Models (LLMs) with increasing levels of required reasoning.

**Process**: To create these 12 variation datasets, we sampled a `Task` for $n$ number of times, where $n$ is proportional to the total count of all Commonly Occurring Configurations of its `Utility Objects`. [Annotation Task 2.1.1] For a given Question's <Task, Utility> Combination, we randomly sample a Context Object from the pool of Context objects. (obtained from 2.1.2). An example of sampling the remaining options is explained below:

**For 5 option datasets**:

1. **Variation-1** : randomly selected Context Object's Ideal Configuration + 4 randomly sampled sub-optimal configurations of the `same` Context Object

2. **Variation-2** : randomly selected Context Object's Ideal Configuration + 2 randomly sampled sub-optimal configurations of the `same` Context Object + 2 randomly sampled sub-

optimal configurations of `different` Context Object belonging to the same <Task,Utility> combination [4]

---

**Example for Task 1 Variation 1**

**Question ID**: 1,
**Utility**: `heating(source)`,
**Question**: `Which of the following objects would be best suited for the purpose of "heating(source)" when tasked to "reheating coffee"?`
**Options**:

- (A) **object name**: `Microwave`, **mass**: `super-heavy`, **temperature**: `RoomTemp`, **material**: `Metal`, **already in use**: `free`, **condition**: `clean`

- (B) **object name**: `StoveBurner`, **mass**: `super-heavy`, **temperature**: `RoomTemp`, **material**: `Metal` **already in use**: `irreversible-using`, **condition**: `dirty`

- (C) **object name**: `CoffeeMachine`, **mass**: `heavy`, **temperature**: `RoomTemp`, **material**: `Metal` **already in use**: `irreversible-using`, **condition**: `dirty`

- (D) **object name**: `Microwave`, **mass**: `super-heavy`, **temperature**: `RoomTemp`, **material**: `Metal` **already in use**: `reversible-using`, **condition**: `clean`

- (E) **object name**: `Microwave`, **mass**: `super-heavy`, **temperature**: `RoomTemp`, **material**: `Metal`, **already in use**: `irreversible-using`, **condition**: `broken`

**Correct Answer**: A

---

**B. Sub-Optimal Configuration Dataset**    Although the process of selecting an ideal configuration is challenging for language models, typically it does not require intricate multi-step reasoning involving consideration of a wide range of factors. To evaluate their reasoning abilities more rigorously (particularly when faced with only sub-optimal options) we now excluded all ideal configurations from our sampling methodology. This deliberate exclusion necessitates that the models engage in more sophisticated reasoning by considering various physical state variables, thereby testing their capacity for abstract reasoning. By focusing exclusively on sub-optimal configurations, this methodological shift enables a more thorough investigation into the language models' ability to navigate and reason through complex scenarios in the absence of clear-cut ideal solutions.

**Process**: To comprehensively assess language models' abstract reasoning capabilities when confronted with sub-optimal configurations, we create another `Variable Level QA dataset` and introduce 14 variations of this dataset. Like the previous dataset, each dataset is constructed using distinct sampling strategies and has a varying number of options. Across all 14 datasets, we maintain a consistent structure of nearly 5,000 questions.

Each question in this dataset variation is associated with a `Task` and `Utility` combination. While the set of questions remains consistent with previous datasets, the sampling of each task is now proportional to the count of "Moderate Configurations + Bad Configurations" (i.e., the count of Sub-Optimal Configurations for that question's associated <Task, Utility> combination). An example of 2 sampling techniques used for generating variation datasets is explained below:

---

[4]The remaining variations in sampling techniques and option count can be found in Appendix B

**For 5 option dataset**

1. **Variation-1**: We sample all 5 options from Moderate Configurations of the Context object of the question's associated <Task,Utility> combination.

2. **Variation-2**: We sample 4 options from Moderate Configurations and 1 option from the Bad Configurations of the Context object of the question's associated <Task,Utility> combination. [5]

---

**Example Task 2 Variation 1**

**Question ID**: 1,
**Utility**: heating(source),
**Question**: Which of the following objects would be best suited for the purpose of "heating(source)" when tasked to "reheating coffee"?
**Options**:

(A) **object name**: CoffeeMachine, **mass**: heavy, **temperature**: RoomTemp, **material**: Metal, **already in use**: reversible-using, **condition**: dirty

(B) **object name**: StoveBurner, **mass**: heavy, **temperature**: RoomTemp, **material**: Metal, **already in use**: reversible-using, **condition**: clean

(C) **object name**: StoveBurner, **mass**: heavy, **temperature**: RoomTemp, **material**: Metal, **already in use**: free, **condition**: dirty

(D) **object name**: CoffeeMachine, **mass**: heavy, **temperature**: RoomTemp, **material**: Metal, **already in use**: reversible-using, **condition**: clean

(E) **object name**: Microwave, **mass**: heavy, **temperature**: RoomTemp, **material**: Metal, **already in use**: reversible-using, **condition**: clean

**Correct Answer**: C

---

## 3 Experimental Setup & Results

Using these 4 datasets, we evaluate various Large Language Models to benchmark their performance on 2 major themes:

1. **R1**: Object-level Commonsense Reasoning (performance on utility and various contextual factors including social constructs, feasibility aspects, etc.)

2. **R2**: Physical State level Commonsense Reasoning (performance on commonsense understanding of various physical variables and how they affect decision making)

We evaluate and compare the performances of various large Language Models using the following metrics:

1. **Accuracy**: The fraction of questions answered correctly by the Language Model.

2. **Bad Rate**: The fraction of questions in which the chosen answer belonged to the "Bad" configuration pool.

---

[5]The remaining variations in sampling techniques and option count can be found in Appendix B

### 3.1 Dataset Summary

| Task | #Variation | #Q | Av | Options | GT |
|:---:|:---:|:---:|:---:|:---:|:---:|
| u | 4 | 2K | 500 | Objects | Utility Objects |
| 0 | 4 | 15.5K | 3.8K | Utility Objects | Context Objects [Õ] |
| 1 | 12 | 58,7K | 4.9K | Õ's Configurations | Õ's Ideal Configurations |
| 2 | 14 | 68.9K | 4.9K | Õ's SuboptConfigurations | Õ's BestSubopt Configurations |

Table 2: Summary of Datasets Used for Experiments

### 3.2 Glossary

| Term | Definition |
|:---|:---|
| *objects* | a set of 100 household objects |
| *utilities* | a set of 22 abstract utilities |
| *tasks* | a set of 75 household activities |
| *question* | a <Task,Utility> combination indicating the specific task aspect to emphasize |
| *variable* | a symbolic variable used to explain an object's physical state |
| *configuration* | complete description of an object using 5 symbolic variables |
| *utility-mapping* | mapping between utility and object; facilitates Utility-Objects |
| *context-mapping* | mapping between task, utility and object; facilitates Context-Objects |
| *ideal-configuration* | A state where an object is ready to perform a task without requiring additional time or effort
marked by the simultaneous occurrence of all ideal variable values in its description |
| *moderate-configuration* | A state requiring additional effort to reach an ideal condition for task performance, marked by the presence of moderate variable values. |
| *bad-configuration* | An inoperative state of an object with irreparable issues, marked by the presence of "Bad Variable" values. |
| *sub-optimal configuration* | group of "Moderate" and "Bad" configurations |

### 3.3 Results

***Task u Analysis***:

| Model | Accuracy | | | |
|:---|:---:|:---:|:---:|:---:|
| | 2-opt | 3-opt | 4-opt | 5-opt |
| PaLM | **98.60** | **96.80** | **96.20** | **92.70** |
| GPT3.5-Turbo | 97.30 | 96.20 | 94.90 | 92.10 |

Table 3: Here, as we move from left to right, the number of options increases in the dataset (from 2 options to 5 options). The findings suggest a near-perfect alignment of objects and their associated utilities. Thus, from here onwards we focus our analysis and discussions on object selection capabilities across contextual factors and physical state variations

---

[5]#Q=question count; Av=average question count

***Task 0 Analysis***: We observe from Table 4 that the performance of GPT3.5-Turbo and PaLM outperform other models with a much smaller number of parameters. This may be attributed to their size as well as the amount of internet data they've been trained on. They both showcased similar performance, suggesting similar object-level reasoning capabilities. Even though the performance of every model was observed to be impressive, Mistral-7B outshone all other models of similar size as well as both 13B models. Upon analyzing the trend of average accuracy across various datasets for Task-0[Figure 3], we note an important trend implying a drop in accuracy as we increase the number of options. This suggests degradation in reasoning capabilities as the number of comparisons increases. This trend was observed in Task 1 and Task 2 as well[6]. Thus through Table 3,4 and Figure 3 we get a fair evaluation of the reasoning capabilities of language models over object-level reasoning tasks. [**R1**]

| Model | Task-0 Variations | | | |
|---|---|---|---|---|
| | 2-opt | 3-opt | 4-opt | 5-opt |
| | **v1** | **v2** | **v3** | **v4** |
| PaLM | **90.0** | **74.2** | **68.3** | 65.0 |
| GPT3.5-Turbo | 88.8 | 71.9 | 68.3 | 66.9 |
| vicuna13B | 71.0 | 54.5 | 49.3 | 46.2 |
| LLama2-13B | **76.5** | **58.2** | **50.9** | **46.9** |
| Vicuna7b | 51.0 | 34.5 | 28.5 | 26.5 |
| Mistral-7B | **76.2** | **57.9** | **50.2** | **47.2** |
| ChatGLM-6B | 62.0 | 42.0 | 34.2 | 27.6 |
| ChatGLM2-6B | **62.9** | **44.6** | **34.3** | **35.4** |

Table 4: Model accuracy when evaluated on Task-0

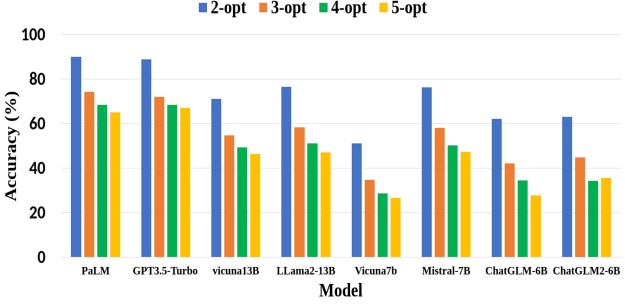

Figure 3: Average Accuracy of various models on Task 0 as we increase option count

***Task 1 Analysis***: Table 5 summarizes the performance accuracy of different models on Task-1 datasets where models were tasked to reason based on physical configuration of objects (using `Ideal Configuration Datasets`). This task was aimed at judging if language models have an understanding of the difference between `Ideal Configuration` and `Sub-Optimal Configurations`. Here also we witness the superior reasoning capabilities of GPT3.5-Turbo and PaLM, with the latter outperforming the former on each dataset by an average of **8.8%**. Amongst the smaller models, we see Mistral7B dominating all other 7B and 6B models. Here Vicuna7B and ChatGLM-6B performed very close to random performances and thus were excluded from further analyses. For 13B models, LLama2-13B showcased its superior reasoning capabilities and was on average **7.6%** more accurate than Vicuna13B. Here, apart from the falling average accuracy with increasing options, we also notice some interesting behaviors as we increase the object diversity (i.e an increase in the number of sub-optimal configurations of `different` context object (of same <Task, `Utility`> combination) barring the object who's Ideal Configuration is already in the options as the correct answer).

---

[6]See Figure 9a, 9b for trends in Task:1 and Task:2 Average Accuracies for each dataset type

Figure 4 illustrates the decreasing performance of all small models as we increase the ob-

| Model | Accuracy for Task_1 Variations⇑ | | | | | | | | | | | |
| --- | --- | --- | --- | --- | --- | --- | --- | --- | --- | --- | --- | --- |
| | 5-option | | | 4-option | | | | 3-option | | | 2-option | |
| | v1 | v2 | v3 | v4 | v5 | v6 | v7 | v8 | v9 | v10 | v11 | v12 |
| PaLM (Chowdhery et al., 2022) | **85.9** | **74.6** | **81.4** | **89.4** | **80.0** | **80.5** | **84.8** | **92.0** | **84.6** | **88.5** | **95.0** | **91.7** |
| GPT3.5-Turbo (Brown et al., 2020) | 74.0 | 64.5 | 70.7 | 78.9 | 69.5 | 69.3 | 75.2 | 89.2 | 78.0 | 82.6 | 91.3 | 80.4 |
| Vicuna13B(Chiang et al., 2023) | 44.5 | 44.0 | 36.4 | 53.2 | 50.9 | 53.3 | 42.9 | 60.6 | 59.9 | 54.1 | 67.8 | 66.6 |
| LLama2-13B(Touvron et al., 2023) | **49.1** | **48.2** | **46.9** | **54.2** | **54.0** | **56.5** | **54.3** | **70.6** | **67.3** | **70.2** | **78.96** | **75.5** |
| Vicuna7B(Chiang et al., 2023) | 24.0 | 24.6 | 24.5 | 31.7 | 32.8 | 32 | 32.2 | 40.5 | 39.2 | 41.0 | 56.8 | 56.8 |
| Mistral-7B(Jiang et al., 2023) | **37.2** | **34.6** | **30.7** | **44.4** | **42.4** | **44.2** | **39** | **53.6** | **51.9** | **47.0** | **74.2** | **69.7** |
| ChatGLM-6B(Du et al., 2022b) | 25.8 | 25.6 | 21.7 | 31.2 | 30.4 | 31.6 | 28.3 | 38.8 | 39.5 | 36.0 | 53.4 | 52.0 |
| ChatGLM2-6B(Du et al., 2022a) | **31.8** | **31.4** | **30.0** | **39.4** | **40.9** | **40.6** | **40.5** | **54.0** | **53.4** | **51.2** | **68.0** | **66.0** |

Table 5: Performance accuracy for various models when evaluated on Task 1 (Ideal Configuration Dataset)

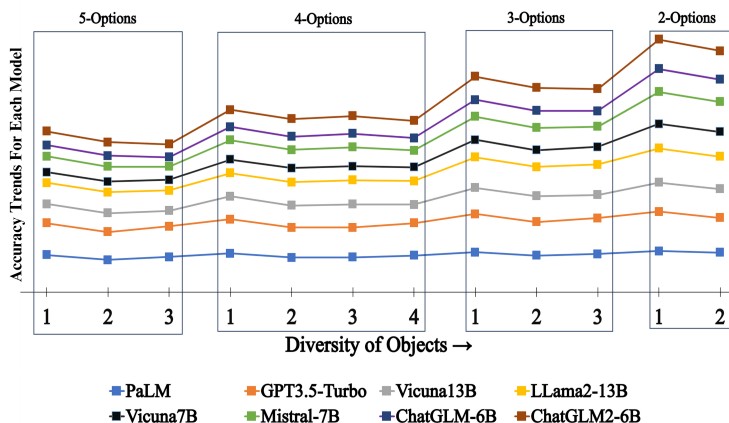

Figure 4: Comparative plot showcasing the variations in Task:1 performances as we keep increasing the **object diversity** in options from left to right.

ject diversity in the options. This sheds light on the existing bias towards using a commonly used object rather than choosing an object decided upon reasoning over every object's complete physical state. However, for big models like PaLM and GPT3.5-Turbo, we notice an improvement in accuracy at extreme object diversities. Thus we conclude that even though there was a drop in accuracy in PaLM and GPT3.5-Turbo for more diverse options – unlike the small models, they were not answering excessively based on their bias towards the commonly used object.

***Task 2 Analysis***: Table 6 summarizes the performance of various models on Task-2, where the models were asked to reason over the best choice of object configurations from the Sub-Optimal Configuration Datasets. This task could be interpreted as finding the option that would be the least time-consuming and most appropriate amongst a variety of Sub-optimal Configurations of

| Model | Accuracy for Task_2 Variations ⇑ | | | | | | | | | | | | | |
|---|---|---|---|---|---|---|---|---|---|---|---|---|---|---|
| | 5-option | | | | | 4-option | | | | 3-option | | | 2-option | |
| | v1 | v2 | v3 | v4 | v5 | v6 | v7 | v8 | v9 | v10 | v11 | v12 | v13 | v14 |
| PaLM | **32.4** | **38.0** | **46.3** | **55.3** | **64.1** | **40.8** | **47.6** | **57.7** | **63.2** | **52.4** | **64.8** | 69.0 | **70.2** | **80.7** |
| GPT3.5-Turbo | 28.3 | 30.6 | 37.5 | 46.4 | 61.6 | 34.6 | 40.1 | 50.72 | 61.2 | 46.1 | 56.7 | **71.3** | 61.1 | 80.2 |
| vicuna13B | 22.5 | 23.9 | 28.0 | 32.0 | 32.8 | 27.7 | 31.0 | 35.3 | 44.2 | 37.3 | 42.9 | 50.0 | 54.8 | 68.4 |
| LLama2-13B | **23.0** | **24.4** | **33.5** | **42.2** | **44.9** | **31.6** | **32.0** | **43.4** | **53.9** | **39.9** | **50.5** | **66.2** | **57.4** | **75.7** |
| Mistral-7B | 20.7 | **22.4** | **27.8** | 25.8 | 27.8 | **25.8** | 29.0 | 32.3 | **37.6** | 35.0 | 40.6 | 47.7 | 52.6 | **63.7** |
| ChatGLM2-6B | **21.6** | 22.2 | 26.5 | **28.2** | **29.0** | 25.6 | **30.6** | **33.6** | 36.3 | **36.5** | **41.9** | **50.7** | **53.7** | 61.4 |
| Vicuna-7B | 20.3 | 21.6 | 21.6 | 21.7 | 22.7 | **26.4** | 25.9 | 27.6 | 28.2 | **33.3** | **35.6** | **38.3** | 48.5 | 50.8 |
| ChatGLM-6B | **21.5** | **22.4** | **22.6** | **23.9** | **23.5** | 25.0 | **27.3** | **29.2** | **29.2** | 33.1 | 34.8 | 36.3 | 48.2 | **53.6** |

Table 6: Accuracy for various models when evaluated on Task 2 (Suboptimal Configuration Dataset)

Context Objects of the question's <Task,Utility> combination. Here, we sampled some moderate configurations (neither Ideal nor Bad) and some Bad Configurations. The best amongst the moderate ones was kept as the Ground Truth. [Refer Appendix D] Our observations reveal consistent superiority of GPT-3.5-Turbo and PaLM across all models. Notably, GPT-3.5-Turbo consistently lags behind PaLM by an average margin of **3.7%**. Despite their commendable comparative performance, both models exhibit limitations in comparing various physical variables of moderate configurations, resulting in a significant performance downturn. Even this time, we observed Vicuna7B and ChatGLM-6B exhibiting erratic behaviors reflected in their consistent random outputs. While LLama2-13B performed superior to all other small-scale models, the general observed order was ChatGLM2-6B ∼ Mistral-7B < Vicuna13B < LLama2-13B < GPT3.5-Turbo < PaLM. In addition to the drop in average accuracy with increasing options, Figure 5 shows the

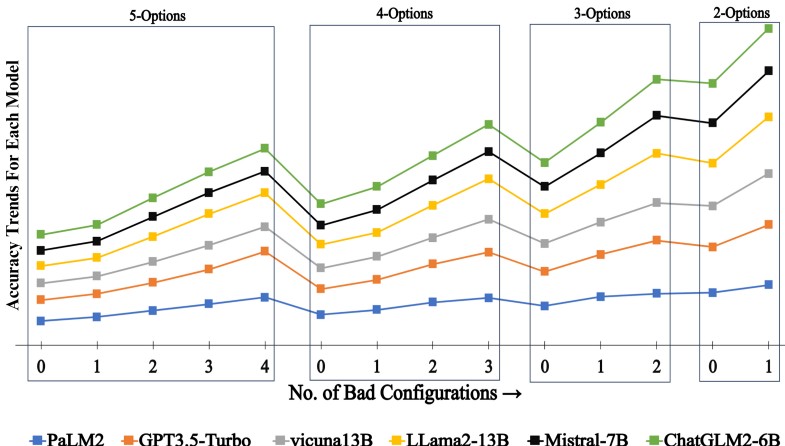

Figure 5: Comparative plot showcasing the variations in Task:2 performances as we keep increasing the **Count of Bad Configurations** in Options from left to right.

trend of enhanced performance as we increase the count of Bad Configurations within a type of dataset. This could be attributed to the ability of models to differentiate Bad Configurations from Moderate Configurations. To delve deeper and analyze what fraction of the responses were correct and what fraction was from the Bad Configurations, we make use of another metric: Bad Rate.

Table 7 shows the percentage of questions where a Bad Configuration was predicted as the correct answer. In our evaluations, this would mean the model went wrong with the reasoning in these questions. To probe LLM reasoning when presented with a varied amount of "bad configurations", we went on to increase the fraction of bad configurations present in each question's options as we moved from left to right([v1→v5], [v6→v9], [v10→v12]). With an increased fraction of options belonging to "bad configurations", we expected an increase in the bad rate. A good model would have a smaller magnitude of the bad rate as well as a larger gap between the fraction of the bad options line (dotted line) and their bad rate value. Figure [6, 7] further showcases the trend of observed bad rates and the trend of the increase of the fraction of bad options we inserted in the prompt. While PaLM and GPT2.5-Turbo showed the least rise in bad rates, we observed LLama2-13B outperforming all other models and consistently trying to achieve PaLM and GPT3.5-Turbo's performance. Based on these figures and analyses, we can safely conclude that most models had a sense of what a bad configuration is but showed limited reasoning capabilities to evaluate moderate configurations based on abstract physical variables. Thus, through Task 1 and Task 2, we were able to evaluate and analyze commonsense reasoning capabilities of language models over physical state variables [**R2**][7]

| Model | Bad Rate For Task_2 Variations ⇓ | | | | | | | | | | | | | |
|---|---|---|---|---|---|---|---|---|---|---|---|---|---|---|
| | 5-option | | | | | 4-option | | | | 3-option | | | 2-option | |
| | v1 | v2 | v3 | v4 | v5 | v6 | v7 | v8 | v9 | v10 | v11 | v12 | v13 | v14 |
| PaLM | - | **4.2** | **10.3** | **21.2** | **35.9** | - | **6.0** | **15.3** | **36.8** | - | **8.8** | 31 | - | **19.3** |
| GPT3.5-Turbo | - | 5.5 | 12.5 | 24.4 | 38.4 | - | 7.7 | 17.5 | 38.8 | - | 9.7 | **28.7** | - | 19.8 |
| vicuna13B | - | 11.2 | 23.2 | 39.0 | 67.2 | - | 15.0 | 33.3 | 55.8 | - | 18.7 | 50 | - | 31.6 |
| LLama2-13B | - | **7.2** | **18.8** | **29.3** | **55.1** | - | **9.6** | **24.6** | **46.1** | - | **13.0** | **33.8** | - | **24.3** |
| Mistral-7B | - | 15.8 | **28.7** | **48.2** | 72.2 | - | **17.0** | 35.7 | **62.4** | - | **19.5** | 52.3 | - | **36.3** |
| ChatGLM2-6B | - | **15.0** | 31.5 | 48.6 | **71.0** | - | **17.0** | **34.6** | 63.7 | - | 20.4 | **49.3** | - | 38.6 |

Table 7: Bad Rate for various models when evaluated on Task 2 (Suboptimal Configuration Dataset). It signifies the fraction of chosen options that were bad choices. (-) is used for datasets where there were no bad options

## 4 Conclusion & Future Work

Accurately reasoning over an object's current physical state is an important aspect of developing robust embodied agents that can accomplish tasks even if the ideal objects are not available. We created a 3 step framework to break the decision-making process that humans go through mentally while choosing an object for task completion. We created 4 major datasets to evaluate object-level and physical state-level reasoning capabilities in LLMs. **Regarding Object Level Reasoning Capabilities(R1)**: Large Language Models were found to have near-perfect object-utility alignment [task-u]. We further found that small models performed decently when evaluated on tasks judging their object-level reasoning capabilities based on contextual factors [task-0]. However, their performance dropped as we increased the number of options in the question. For large LLMs like PaLM and GPT3.5-Turbo, we observed very impressive object-level selection capabilities. However, here as well we saw a drop in accuracy as we increased the number of options in each question.

---

[7]For additional experiments and fine-tuning results see Appendix F.1

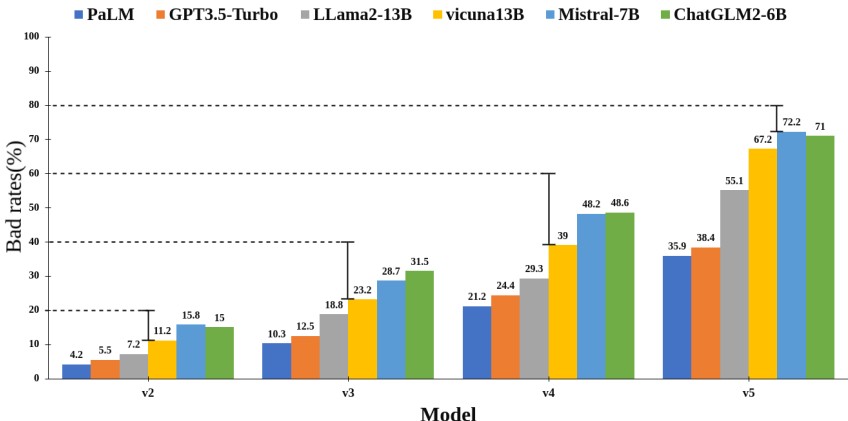

Figure 6: Bad rates for various 5 option datasets as we increase the count of bad options from left to right. Dotted Lines represent the fraction of options that were bad in each dataset, whereas the difference between the dotted lines and bars tells us about the model's ability to not get confused as we increase the count of bad options from left to right on the x-axis

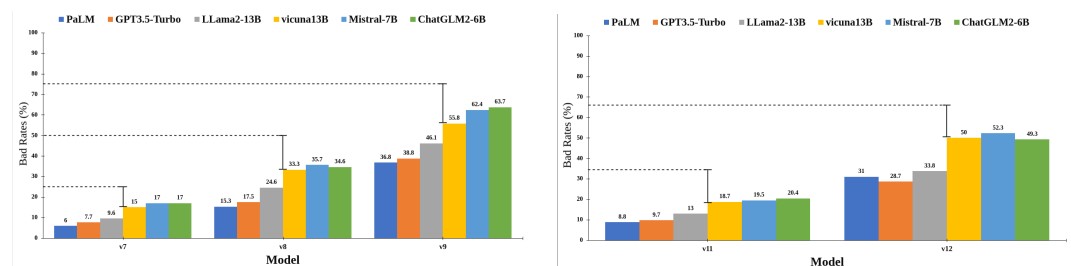

Figure 7: Bad rates for various `4(left)` and `3(right)` option datasets as we increase the count of bad options. While the dotted line represents the fractions of options that were bad in each dataset, the difference between the dotted lines and bars depicts the model's ability to not get confused as we increase the count of bad options from left to right on the x-axis

**Regarding Physical State-Level Reasoning Capabilities(R2)**: While evaluating commonsense reasoning over an object's physical state, we noticed that all language models displayed impressive abilities to identify `Ideal Configurations` on task-1. For small models, we noticed a decreasing accuracy within each fixed count option dataset(task-1) as we increased the object diversity. This brought forth the internal bias of these small models to stick to an object commonly used for a task, even if it is not in an Ideal State or a condition to be readily used. However, for larger models, we observed a lesser degradation in such accuracy. There was a stark difference in how large and small LLMs behaved when tasked to choose the most appropriate moderate configuration. While all language models displayed an increase in accuracy as we increased the fraction of `Bad Configurations` in each question (task-2), not all were able to avoid confusion between **Moderate Configurations** and **Bad Configurations**. This ability to identify and choose Moderate Configurations amongst a mixture of Bad and Moderate Configurations decreased as we decreased the size of LLMs. (as

shown by bad rate plots). Similar to task-u, task-0, and even in task-1 and task-2, we observed a decrease in accuracy rates as we kept increasing the number of options in each question.

Given these observations, we can safely conclude that language models like GPT3.5-Turbo, PaLM, and Llama2-13B can prune out appropriate objects based on utility[Task-u] and contextual factors[Task 0] and extreme physical configurations (Ideal and Bad Configurations) to an impressive extent. However, they face a certain level of difficulty in comparing amongst Moderate Configurations (as this requires a certain amount of abstract reasoning equipped with a commonsense understanding of the world around them) **[R2]**. Smaller language models showcase sub-optimal behavior over **[R1]** and very poor behavior over **[R2]**

Our work opens up an avenue for improving the language model's abstract multi-step reasoning for estimating the physical affordance of everyday objects used in household activities. Future efforts would be directed towards integrating these datasets to train Embodied Language agents and proving their competence of our 3-step architecture in successful task completion when situations aren't ideal. Judging the variable values in the real world could be a tricky affair; thus, although the current work focused on handcrafted variables, calculating these variables and learning new latent variables from multi-modal inputs for effective analysis and reasoning about an object's applicability seems a foreseeable domain to explore.

## Limitations

This work focuses on dealing with contextual connotations associated with an object when deciding whether to use it as a substitute for task execution. We further considered abstract physical variable level analysis to highlight the evolution of usability with various physical abstractions. While determining the values of these variables may appear straightforward in the AI2Thor Simulator, achieving the same in real-life scenarios requires a resilient model. Even if we can calculate the variables, there is a limitation to which an object's state could be represented using abstract physical variables. When comparing objects, sometimes we need to understand their exact situation to decide their usability. To develop robust embodied agents capable of dealing with such explicit reasoning along with abstract commonsense reasoning capabilities, further work needs to be directed along with integrating multi-modal reasoning capabilities in addition to commonsense reasoning. In addition, in this study, we assumed that all the objects were allowed to be used by the agent. In some cases, it might be possible that the human companion of the agent might have kept an object in a certain way and didn't want it disturbed. Thus, the agent might need to re-calculate the object use preference as per this newly imposed human preference. Further works along this line would enable us to move an inch closer toward Embodied agents capable of such constrained planning capabilities in addition to multi-modal commonsense reasoning.

## 5   Related Works

A lot of work has been done in the domains related to the scope of this paper. In this section, we summarize some of them:

**Probing Language Models**  Understanding what LMs know after large-scale pre-training is an active research area (Rogers et al., 2020). Various probing methods have been developed (Tenney et al., 2019b); (Petroni et al., 2019), and investigations show that LMs capture linguistic (Tenney et al., 2019a); (Liu et al., 2019), factual (Petroni et al., 2019); (Roberts et al., 2020); (Dai et al., 2022), commonsense knowledge (Wang et al., 2019); (Forbes et al., 2019), and even acquire grounded concepts (Patel & Pavlick, 2021).

**CommonSense QA Datasets**  Evaluating to what level commonsense world understanding LMs possess has been explored by many. (Gu et al., 2023) analyses mental models of LLMs and aligns them with improved models about everyday things; (Bisk et al., 2019) consisted of questions requiring physical commonsense reasoning. Recently, there has been a lot of work in NLP to utilize commonsense for QA, NLI, etc. (Sap et al., 2019); (Talmor et al., 2019). Many of these approaches seek to effectively utilize ConceptNet by reducing the noise retrieved from it (Lin et al., 2019) (Kapanipathi et al., 2020) There have been several other QA Datasets to benchmark CommonSense Reasoning abilities in Language Models. Some of them include: (Geva et al., 2021); (Yang et al., 2018); (Mihaylov et al., 2018);

**Reasoning in LLMs**  Reasoning is a crucial aspect of intelligence, influencing decision-making, problem-solving, and other cognitive abilities. (Huang & Chang, 2023) presents the current state of research on LLMs' reasoning abilities, exploring approaches to improve and evaluate their reasoning skills. (Dziri et al., 2023) investigates problems associated with multistep reasoning with LLMs Some of the works dealing with tackling reasoning in small models are: (Magister et al., 2023) (Fu et al., 2023) (Shridhar et al., 2023)

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

# A Appendix

## A.1 Dataset Specifics

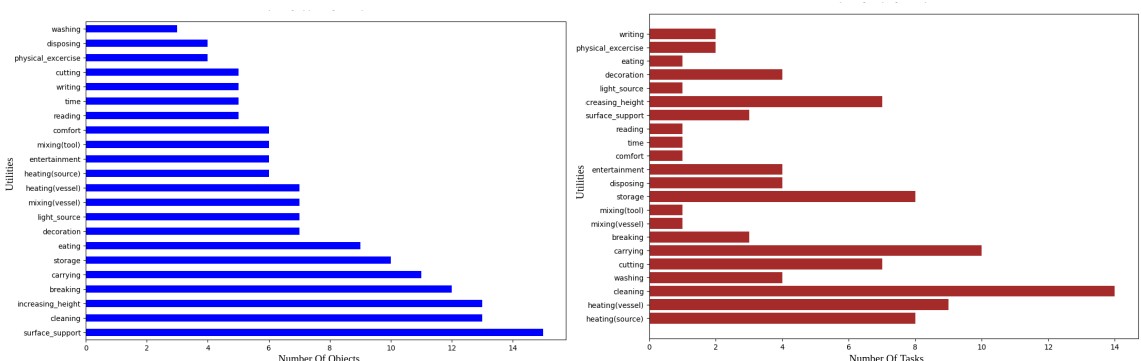

(a) Plot showing number of objects(x) for each utility(y), as obtained after Utility based pruning 2.1.1

(b) Plot showing number of tasks(x) for each utility(y)

## A.2 Variables

Here we describe the variables used to describe an object's physical state. We kept it at an abstract level to judge basic commonsense reasoning capabilities.

1. **mass**: Based on an estimate of the weight of an object: **(i)** light[0,1 Kg] **(ii)** medium[1,5 Kg] **(iii)** heavy[5,10 Kg] **(iv)** super-heavy[> 10 Kg]

2. **material**: what material is used to make that object

3. **temperature**: the surface temperature of the object: Cold/Hot/RoomTemp

4. **already in use**: tells us about the availability of the object: reversible-using/irreversible-using/free

5. **condition**: tells us about the condition of the object: broken/clean/dirty

## A.3 Human Annotations: Object-Utility Mappings

Table 2.1.1 summarizes the collected and refined Object-Utility pairings. Throughout this work, we have referred to these as `Utility Mappings`.

# B Dataset Creation

In addition to the variations explained in 2.2.3, we further create 3 more types of datasets for each of the 4 tasks. These would be each consisting of 4, 3, and 2 options. Our sampling method for these options enables us to analyze and ablate over reasoning capabilities of LLMs in a zero-shot manner. The datasets are:

### B.1 Task 0

1. **Variation-2**: For each question, we sampled 1 context object and 2 utility objects belonging to the same utility.

2. **Variation-3**: For each question, we sampled 1 context object and 3 utility objects belonging to the same utility.

3. **Variation-4**: For each question, we sampled 1 context object and 4 utility objects belonging to the same utility.

### B.2 Task 1

**5 option datasets** :

1. **Variation-3**: Random context object's Ideal Configuration + 4 randomly sampled sub-optimal configurations of same Task and Utility's different context object

**4 option datasets** :

1. **Variation-4** : Random context object's Ideal Configuration + 3 randomly sampled sub-optimal configurations of the same context object

2. **Variation-5**: Random context Object's Ideal Configuration + 2 randomly sampled sub-optimal configurations of the same context object + 1 randomly sampled sub-optimal configurations of different context object belonging to the same <Task,Utility> combination

3. **Variation-6**: Random context Object's Ideal Configuration + 1 randomly sampled sub-optimal configurations of the same context object + 2 randomly sampled sub-optimal configurations of different context object belonging to the same <Task,Utility> combination

4. **Variation-7**: Random context object's Ideal Configuration + 3 randomly sampled sub-optimal configurations of the different context object belonging to the same <Task,Utility> combination

**3 option datasets** :

1. **Variation-8** : Random context object's Ideal Configuration + 2 randomly sampled sub-optimal configurations of the same context object

2. **Variation-9**: Random context object's Ideal Configuration + 1 randomly sampled sub-optimal configuration of the same context object + 1 randomly sampled sub-optimal configuration of different context object belonging to the same <Task,Utility> combination

3. **Variation-10**: Random context object's Ideal Configuration + 2 randomly sampled sub-optimal configurations of the different context object belonging to the same <Task,Utility> combination

**2 option dataset**

1. **Variation-11**: Random context object's Ideal Configuration + 1 randomly sampled suboptimal configurations of the same context object.

2. **Variation-12**: Random context object's Ideal Configuration + 1 randomly sampled suboptimal configurations of the different context object belonging to the same <Task,Utility> combination

## B.3   Task 2

**5 option dataset**

1. **Variation-3**: We sample 3 options from the Moderate Configurations and 2 options from the Bad Configurations of the same <Task, Utility> combination's context objects
2. **Variation-4**: We sample 2 options from the Moderate Configurations and 3 options from the Bad Configurations of the same <Task, Utility> combination's context objects

3. **Variation-5**: We sample 1 option from the Moderate Configurations of the context objects of that particular <Task, Utility> combination, we allow sampling equivalent options as long as either of them is not the correct answer. We also sample 4 options from the Bad Configurations of context objects of that particular <Task, Utility> combination

**4 option dataset**

1. **Variation-6**: We sample 4 options from the Moderate Configurations of the context objects of that particular <Task, Utility> combination. Here we allow sampling equivalent options as long as either is not the correct answer.

2. **Variation-7**: We sample 3 options from the Moderate Configurations of the context object of that particular <Task,Utility> combination. We allow sampling equivalent options as long as either is not the correct answer. We also sample 1 option from the Bad Configurations of the random context objects of that particular <Task, Utility> combination

3. **Variation-8**: We sample 2 options from the Moderate Configurations of the context object of that particular <Task,Utility> combination. We allow sampling equivalent options as long as either is not the correct answer. We also sample 2 options from the Bad Configurations of the random context objects of that particular <Task, Utility> combination

4. **Variation-9**: We sample 1 option from the Moderate Configurations of the context objects of that particular <Task, Utility> combination. Here, we allow sampling equivalent options as long as either is not the correct answer. We also sample 3 options from the Bad Configurations of context objects of that particular <Task, Utility> combination

**3 option dataset**

1. **Variation-10**: We sample 3 options from the Moderate Configurations of the context objects of that particular <Task, Utility> combination. Here, we allow sampling equivalent options as long as either is not the correct answer.

2. **Variation-11**: We sample 2 options from the Moderate Configurations of the context objects of that particular <Task, Utility> combination. Here, we allow sampling equivalent options as long as either is not the correct answer. We also sample 1 option from the Bad Configurations of context objects of that particular <Task,Utility> combination

3. **Variation-12**: We sample 1 option from the Moderate Configurations of the context object of that particular <Task,Utility> combination. We also sample 2 options from the Bad Configurations of the random context objects of that particular <Task, Utility> combination

**2 option dataset**

1. **Variation-13**: We sample 2 options from the Moderate Configurations of the context objects of that particular <Task, Utility> combination, here we allow sampling equivalent options as long as either of them is not the correct answer.

2. **Variation-14**: We sample 1 option from the Moderate Configurations of the context object of that particular <Task,Utility> combination. We also sample 1 option from the Bad Configurations of the random context objects of that particular <Task, Utility> combination

## C Results

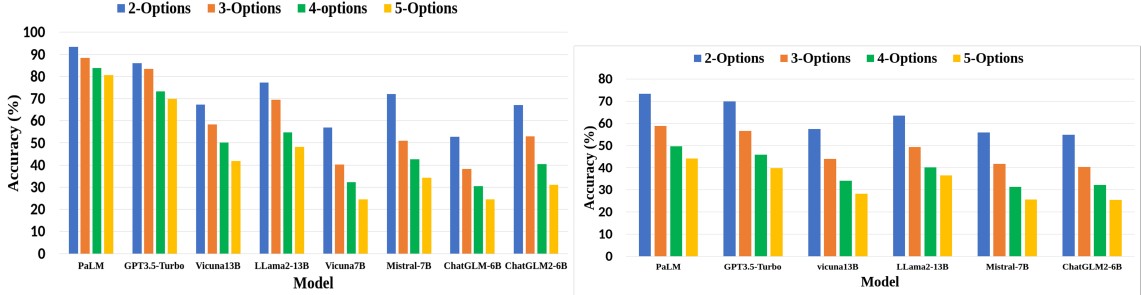

(a) Average Accuracy of Various models on Task 1 as we increase option count

(b) Average Accuracy of Various models on Task 2 as we increase option count

## D Annotation Process

The entire annotation process was text-based and was executed by circulating a text-based questionnaire. Participant demographic spanned various university-level academic departments and consisted of students and researchers who volunteered for such annotations. Figure 10 summarizes the entire annotation process for generating Ground Truths for all 4 datasets.

### D.1 Human Annotations: Utility-Object Mappings

Creation of utility-object mappings that were further used as the backbone for all the tasks and datasets involved the use of GPT3.5-Turbo and Human Annotation. This was done by using

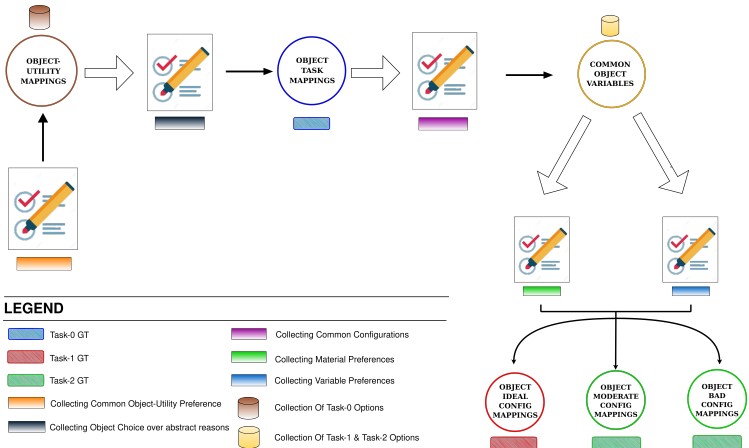

Figure 10: Figure summarizing our annotation process

GPT3.5-Turbo to output Utilities for the 100 selected AI2Thor objects. From this, we then selected a random subset (after cross-checking it) and used it to create options while generating QnA to gather human annotation for Utility-Object Mappings. The annotators were asked to label 100 objects with utilities from a list of 22 utilities. The inter-annotator agreement was calculated by formulating this as a multi-annotator-multi-label scenario where each annotator could annotate a variable number of labels per object. The annotator agreement was **89.2%**, suggesting a high degree of agreement within the annotators. The consolidated utility-object mappings are found Link

### D.2 Human Annotations: Task-Object Mappings

To curate ground truth task-object mappings, also called `Context Mappings`; we ask the annotators to choose objects appropriate for a $<$`Task`, `Utility`$>$ combination amongst the utility objects. As one question can have more than 1 possible correct object, we calculated inter-annotator agreement by modeling this as a process similar to the previous annotation task. The annotator agreement was observed to be: **81.0%**, suggesting a high degree of agreement amongst the annotators. The question posed to the annotators was similar to the ones used to curate Task 0 (Object Level Dataset), and the obtained responses were used as Ground Truth for Task 0 Dataset. The processed GT can be found here: Link

### D.3 Human Annotations: Common Object-Variables Mappings

To get the common variable values for all the objects, we further ask the annotators to provide all commonly occurring variable values of each object. Using these, we created all possible configurations. Upon calculating the inter-annotator agreement as earlier annotation tasks, we observed an inter-annotator agreement of **89.9** when averaged across all 5 variables. The processed output can be found here: Link

### D.4 Human Annotations: Ideal Object Configurations

Further, we ask the annotators to categorize variable values into 3 categories: Ideal, Moderate, and Bad. "Ideal" refers to an ideal state of the object; "moderate" means you have to spend some time getting the object in an ideal state before it can be used, whereas "bad" means the object is unusable. Some variable values are obvious, such as "free", which would be ideal, whereas; "reversible-using" would be moderate, and "irreversible-using" would be bad. So we only ask them to give preference for variables like Material. The observed Krippendorff's reliability alpha (Krippendorff, 2011) among the raters for classifying material variable values into categorical variables: "Ideal", "Moderate", and "Bad" was **0.87**, suggesting a high degree of agreement amongst the annotators. The Ideal Configurations can be found Link.

### D.5 Human Annotations: Moderate Configurations

After classifying the variable values into these 3 categories, we asked them to arrange the values in increasing order of their appropriateness for a given <Task,Utility> combination. For ranks as ordinal variables, we observed Krippendorff's alpha value to be **0.89**, showing a high agreement amongst the annotators. Further, we set a penalty for each moderate variable value and consequently generate 2 penalty scores for each configuration: material penalty and time penalty. It is using these penalties we further arrange the configurations based on a time penalty and then a material penalty. This helps us create a relative ranking within the moderate configurations and enables us to sample "moderate" options when curating Task 2 Dataset.

### D.6 Human Annotations: Bad Configurations

For the Bad Configurations, we set abnormally high values for material and time penalties. These configurations help us sample "bad" options when curating Task 2 Dataset. The sub-optimal configurations, including "moderate" and "bad" configurations, can be found here: Link

## E Prompts used

Prompts used for various models can be found at this link. Link

## F    Example Responses

---

**Task 0**

**Question ID**: 1,
**Utility**: `heating(source)`,
**Question**: `Which of the following objects would be best suited for the purpose of`
`"heating(source)" when tasked to "reheating coffee"?`
**Options**:

   (A) **object name**: `Toaster`,

   (B) **object name**: `StoveBurner`

**Correct Answer**: B

---

**PaLM**

**Answer**: `B`
**Reason**: Stove burners are safe and feasible, require minimal effort, and reheat coffee in a minimal amount of time.

---

**GPT3.5-Turbo**

**Answer**: `B`
**Reason**: Option B, StoveBurner, would be best suited for the purpose of reheating coffee in terms of safety, feasibility, minimum effort, and minimum time. Using a stove burner allows for precise control of heat, enabling a gradual and controlled reheating of the coffee. It's a common method for reheating liquids and is safer than using a toaster, which is not designed for reheating liquids and could pose a safety risk or result in spillage or damage to the toaster.

---

**LLama2-13B**

**Answer**: `A`
**Reason**: Toaster is the best option for reheating coffee as it is designed specifically for heating bread and other toastable items, and it can heat up quickly and evenly. Using a stove burner would require more effort and time to heat the coffee, and it may not heat it evenly. Additionally, using a toaster is generally safer than using an open flame on a stove burner, as it has built-in safety features such as a cool-touch exterior and automatic shut-off.

---

**Vicuna7B**

**Answer**: `B`
**Reason**: Option B, a stove burner, is the best choice for reheating coffee as it is a safe and efficient heating source that requires minimal effort and time. It allows for precise temperature control and can be easily used to reheat coffee without causing any damage to the coffee or creating any safety hazards.

## Vicuna-13B

**Answer**: B
**Reason**: A stove burner would be the best option for reheating coffee as it provides precise temperature control, allowing for easy adjustment to the desired temperature. This ensures that the coffee is heated evenly and efficiently, without the risk of burning or overheating. Additionally, a stove burner is a convenient and easily accessible option, requiring minimal effort and time to use. On the other hand, a toaster may not be the best option as it is not designed for heating liquids and may not provide consistent or accurate temperature control.

## Mistral7B

**Answer**: B
**Reason**: While a toaster is a convenient option, a stove burner would be the best suited for heating coffee as it provides more control over the temperature and the amount of heat being applied. It also allows for a larger surface area to be heated, which can be beneficial in reheating coffee. Additionally, using a stove burner is safer and more energy efficient than using a toaster

## ChatGLM-6B

**Answer**: A
**Reason**: Toaster is a more convenient and efficient option for reheating coffee compared to a stove burner.

## ChatGLM2-6B

**Answer**: A
**Reason**: Toaster is a device that heats up food items such as coffee. It is a more convenient and safer option compared to a stove burner. The toaster has a simple operation, low maintenance and can be used to heat up a single serving of coffee at a time. The time it takes to heat up the coffee is minimal and there is no need for additional effort to operate it.

## Task 1

**Question ID**: 1,
**Utility**: heating(source),
**Question**: Which of the following objects would be best suited for the purpose of "heating(source)" when tasked to "reheating coffee"?,
**Options**:

(A) **object name**: Microwave, **mass**: super-heavy, **temperature**: RoomTemp, **material**: Metal, **already in use**: free, **condition**: clean

(B) **object name**: StoveBurner, **mass**: super-heavy, **temperature**: RoomTemp, **material**: Metal **already in use**: irreversible-using, **condition**: dirty

(C) **object name**: CoffeeMachine, **mass**: heavy, **temperature**: RoomTemp, **material**: Metal **already in use**: irreversible-using, **condition**: dirty

(D) **object name**: Microwave, **mass**: super-heavy, **temperature**: RoomTemp, **material**: Metal **already in use**: reversible-using, **condition**: clean

(E) **object name**: Microwave, **mass**: super-heavy, **temperature**: RoomTemp, **material**: Metal, **already in use**: irreversible-using, **condition**: broken

**Correct Answer**: A

**PaLM**

**Answer**: A
**Reason**: Microwaves are safe and easy to use, and they can heat coffee quickly and evenly. They are also relatively lightweight and easy to move, so they can be used in a variety of settings.

**GPT3.5-Turbo**

**Answer**: D
**Reason**: Microwave (option D) is the best choice as it is available (reversible-using), in clean condition, and suitable for reheating coffee. It is not irreversibly occupied and can be used for the task without a time penalty.

**LLama2-13B**

**Answer**: A
**Reason**: Microwaves are designed specifically for heating and cooking, making them a safer choice compared to stovetop burners or coffee machines. Additionally, microwaves are generally made of durable materials and have a more delicate design, which makes them less fragile than stovetop burners or coffee machines. Furthermore, microwaves are quick and efficient when it comes to heating up liquids, taking less time than stovetop burners or coffee machines. Finally, option A has already been used (already in use: "free") and its condition is clean, which suggests that it is ready for immediate use.

**Vicuna7B**

**Answer**: D
**Reason**: This is because a microwave is a common and convenient appliance for reheating coffee quickly and efficiently, and it can be used multiple times without any issues. Additionally, a microwave is generally safe and easy to use, and it does not require any time penalty or additional setup to get started with the task.

**Vicuna13B**

**Answer**: A
**Reason**: A microwave is the best suited object for the purpose of reheating coffee as it is a fast and efficient way to heat liquids. It is also a reversible use object as it can be used again after the task is completed. Additionally, a microwave is a safe and convenient option as it eliminates the need for hot surfaces or open flames, which can be hazardous. The mass and material properties of a microwave also make it a safe and practical choice for this task.

---

**Mistral-7B**

**Answer**: A
**Reason**: Based on the given physical variables, object A (Microwave) would be the best suited for the purpose of heating(source) when tasked to reheating coffee. The microwave is readily available and clean, which means it can be used immediately without any additional time required for cleaning or preparation. Additionally, microwaves are generally safe for use in heating food and are less likely to break down compared to other materials. The microwave temperature is also at a safe level for heating food. While other options such as stove burners and coffee machines may also be suitable for heating coffee, they may require additional time and effort to clean and prepare for use, and their availability may be limited due to current use or maintenance requirements. Therefore, the microwave is the best choice for reheating coffee in a household setting.

---

**ChatGLM-6B**

**Answer**: D
**Reason**: The best option for reheating coffee would be the Microwave object with id 5, which has already been mentioned to be reversible and in use, but also has a lower mass compared to other options, making it easier to handle and transport. Additionally, it is a metal microwave, which can provide better heating efficiency compared to other options made of different materials.

---

**ChatGLM2-6B**

**Answer**: B
**Reason**: The stove burner is suitable for heating coffee as it has a high temperature and is made of metal, which is conductive and can efficiently transfer heat. Additionally, it is in a state of reversible using, which means it is temporarily unavailable but can be used after some time. The condition is dirty, which is not relevant to the task of reheating coffee.

---

**Task 2**

**Question ID**: 1,
**Utility**: `heating(source)`,
**Question**: `Which of the following objects would be best suited for the purpose of "heating(source)"` `when tasked to "reheating coffee"?`,
**Options**:

(A) **object name**: `CoffeeMachine`, **mass**: `heavy`, **temperature**: `RoomTemp`, **material**: `Metal`, **already in use**: `reversible-using`, **condition**: `dirty`

(B) **object name**: `CoffeeMachine`, **mass**: `heavy`, **temperature**: `RoomTemp`, **material**: `Metal`, **already in use**: `irreversible-using`, **condition**: `clean`

(C) **object name**: `CoffeeMachine`, **mass**: `heavy`, **temperature**: `RoomTemp`, **material**: `Metal`, **already in use**: `free`, **condition**: `dirty`

(D) **object name**:`Microwave`, **mass**: `super-heavy`, **temperature**: `RoomTemp`, **material**: `Metal`, **already in use**: `reversible-using`, **condition**: `dirty`

(E) **object name**:`StoveBurner`, **mass**: `super-heavy`, **temperature**: `RoomTemp`, **material**: `Metal`, **already in use**: `irreversible-using`, **condition**: `clean`

**Correct Answer**: C

---

**PaLM**

**Answer**: `A`
**Reason**: The coffee machine is the most efficient way to reheat coffee, and it is also the safest option.

---

**GPT3.5-Turbo**

**Answer**: `E`
**Reason**: StoveBurner is available, clean, and can be used without time penalty (reversible-using). It is suitable for heating and reheating tasks and is generally quick to heat up.

---

**LLama2-13B**

**Answer**: `C`
**Reason**: The Coffee Machine is the best configuration for reheating coffee because it is the least time-consuming and most reliable option. Although it is currently dirty, it can be cleaned after use, making it a more reliable choice than the Microwave or StoveBurner, which may have permanent damage if used with dirty or wet objects. Additionally, the Coffee Machine is already available and does not require any time to "get started" like the Microwave or StoveBurner, which need time to heat up. Therefore, the Coffee Machine is the best option for reheating coffee based on commonsense reasoning related to the provided physical variables.

---

**Vicuna-7B**

**Answer**: `C`
**Reason**: The Coffee Machine is the best option for the purpose of "heating(source)" when tasked to "reheating coffee" because it is already available and not in use, and it is also in a dirty condition. Therefore, it is the most suitable option to use for this task.

**Vicuna-13B**

**Answer**: A
**Reason**: The best option for reheating coffee is the 'Coffee Machine' because it is specifically designed for making and heating coffee. It has a 'reversible-using' availability status, meaning it can be used after some time. It is also made of metal, which is a safe and durable material for heating purposes. Additionally, it is already dirty, which means it can be cleaned after use without any extra time penalty. Overall, the 'Coffee Machine' is the most suitable option for reheating coffee.

**Mistral-7B**

**Answer**: A
**Reason**: The Coffee Machine with already in use = reversible using and condition = dirty is the best option for reheating coffee. It is readily available and can be easily cleaned before use, making it safe and efficient

**ChatGLM-6B**

**Answer**: A
**Reason**: The Coffee Machine is the best suited for the purpose of heating(source) when tasked to reheating coffee, as it is lightweight, easy to handle and can be used multiple times without any loss of heat. Additionally, it is metal and can be easily heated to a high temperature, making it suitable for heating coffee.

**ChatGLM2-6B**

**Answer**: C
**Reason**: The Coffee Machine has a free state which means it can be readily used without time penalty. Additionally, it has a dirty condition which is temporary and can be overcome. It is also made of metal which may not be safe for food contact. It is also a heavy object which may be difficult to move.

| Utility | Objects |
|---|---|
| Heating (source) | Microwave, Stove Burner, Toaster, Oven, Coffee Machine, Candle |
| Carrying | Wine Bottle, Watering Can, Vase, Bottle, Cup, Mug, Pot, Kettle, Bowl, Plate, Spray Bottle |
| Cleaning | Dish Sponge, Scrub Brush, Plunger, Paper Towel Roll, Soap Bar, Soap Bottle, Towel, Newspaper, Toilet Paper, Cloth, Hand Towel, Tissue Box, Vacuum Cleaner |
| Washing | Bathtub Basin, Sink Basin, Toilet |
| Cutting | Butter Knife, Knife, Fork, Spatula, Spoon |
| Disposing | Garbage Can, Bathtub Basin, Sink Basin, Toilet |
| Mixing (tool) | Ladle, Fork, Spoon, Spatula, Butter Knife, Knife |
| Mixing (vessel) | Pan, Bowl, Kettle, Cup, Plate, Pot, Mug |
| Heating (vessel) | Pan, Bowl, Kettle, Cup, Plate, Pot, Mug |
| Storage | Drawer, Cabinet, Dresser, Fridge, Laundry Hamper, Safe, Shelf, Shelving Unit, Desk, Box |
| Entertainment | Television, CD, Remote Control, Cell Phone, Laptop, Desktop |
| Comfort | Bed, Armchair, Chair, Dog Bed, Sofa, Ottoman |
| Reading | Book, Newspaper, Desktop, Laptop, Cell Phone |
| Increasing Height | Desk, Armchair, Chair, Coffee Table, Footstool, Dining Table, Countertop, Stool, Ottoman, Sofa, Side Table, Dresser, Bed |
| Time | Watch, Alarm Clock, Cell Phone, Desktop, Laptop |
| Eating | Apple, Bread, Potato, Lettuce, Egg, Coffee, Tomato, Salt |
| Physical Exercise | Basketball, Baseball Bat, Dumbell, Tennis Racket |
| Writing | Pen, Pencil, Cell Phone, Laptop, Desktop |
| Surface Support | Coffee Table, Countertop, Desk, Dining Table, Dog Bed, Dresser, Floor, Footstool, Ottoman, Side Table, Sofa, Stool, Chair, Armchair, Bed |
| Light Source | Light Switch, Window, Desk Lamp, Blinds, Curtains, Candle, Floor Lamp |
| Decoration | Statue, House Plant, Room Decor, Teddy Bear, Tabletop Decor, Poster, Painting |
| Breaking | Butter Knife, Knife, Fork, Spatula, Plate, Ladle, Basketball, Tennis Racket, Dumbbell, Remote Control, Baseball Bat, Spoon |

Table 8: Utilities and Objects

### F.1    Fine-tuning Results

Upon fine-tuning a language model with a subset of various datasets that we curated, we expected to see an increase in the accuracy of the model. Below, we present the results obtained after we fine-tuned a PaLM model on Vertex AI.

#### F.1.1    Task-0 Fine-tuning: Model for Object Level Selection

Due to the limitation of computational resources, we selected a slice of 400 examples of **5-option variation dataset** and fine-tuned the PaLM language model for 40 training steps. In Table 9, we present the comparison of the results before fine-tuning and after such minimal fine-tuning. Owing to the increase in accuracy across all variations after fine-tuning just 450 examples of the 5-option datasets for 40 training steps, we can safely expect a substantial increase when fine-tuned on a larger split of datasets for a larger number of training steps.

#### F.1.2    Task-1,2 Fine-tuning: Model for Physical State Level Selection

Due to the limitation of computational resources, we selected a slice of 1200 examples which included 450 examples from all 3 variations of task-1's **5-option variation dataset**[see Table 5] and 750 examples from all 5 variations of task-2's **5-option variation dataset**[see Table 6]. We further fine-tuned a single PaLM language model for 40 training steps. Table 10 presents the result comparison before fine-tuning and after such minimal fine-tuning.

We note some common observations observed after fine-tuning both these models:

1. Even with fine-tuning them on a small subset of 5 variation datasets, we got an increase in accuracy in all datasets.

2. We got impressive results with minimal fine-tuning for 40 training_steps. We can safely expect a substantial increase when fine-tuned on a larger split of datasets for a larger number of training steps.

### F.2    Full Pipeline Evaluations

> **Comments on Previous Experiments**
>
> Previously, we designed (task-u, task-0) and (task-1, task-2) to evaluate object level and physical state level choosing capabilities, respectively. Here we ensured that all other factors were kept ideal while evaluating a specific factor. For example, while evaluating object-level selection capabilities, we didn't specify the physical state and instructed the model to assume them in perfectly usable conditions. Whereas, while evaluating physical state level-based selection capabilities, we only provided the context (best suitable) objects corresponding to the `<utility, task>` combination, thus eliminating any errors arising from selecting the wrong objects. These were designed to evaluate object-level reasoning and physical state-level reasoning abilities of LLMs **individually**.

To evaluate the performance of language models when tasked to employ both these reasoning abilities (object level and physical state level), we designed 2 new datasets consisting of options where either the object could be inappropriate, the physical state could be inappropriate, or both.

| Model | Obj. Level Accuracy | | | |
|-------|-------|-------|-------|-------|
|       | 2-opt | 3-opt | 4-opt | 5-opt |
| PaLM | 90.0 | 74.2 | 68.3 | 65.0 |
| PaLM FT | **91.5** | **75.6** | **70.9** | **68.2** |

| Model | Physical Level Accuracy | | | | | | | |
|-------|------|------|------|------|------|------|------|------|
|       | 2-opt | | 3-opt | | 4-opt | | 5-opt | |
|       | t1 | t2 | t1 | t2 | t1 | t2 | t1 | t2 |
| PaLM | 92.5 | 86.9 | 88.7 | 75.2 | 80.4 | 66.9 | 71.3 | 60.4 |
| PaLM FT | **99.2** | **88.8** | **94.9** | **79.3** | **92.5** | **72.6** | **89.3** | **66.0** |

Table 9: [**Task-0**]Here we compare the average accuracy of the PaLM language model before and after fine-tuning when evaluated on various types of fixed option-count datasets for task-0 (as in Table 4). We can observe a substantial increase in accuracy for task-0 performance even by fine-tuning on such a small subset of our data (400 examples for just 40 training_steps)

Table 10: [**Task-1,2**]Here, we compare the average accuracy of the PaLM language model before and after fine-tuning when evaluated on 500 questions of each variation of task-1 and task-2 dataset. (from Table 5, 6)from various types of fixed-count datasets. For each value, we averaged the accuracy of all variations that were a part of that fixed count dataset. We observed a substantial increase in accuracy for task-1 and task-2 performances even after fine-tuning on a small subset of data (1200 examples with task-1:task-2 ratio was 3:5) for just 40 training_steps.

### F.2.1 Full$_{ideal}$ Dataset

In this dataset, the correct answer would be the `ideal configuration` of the `context object`. Meanwhile, the other present options could include sub-optimal configurations of context objects, any configurations (ideal, sub-optimal) of utility, and any unrelated random object. We created around 30 variations of 15K QnA Pairs with varying option counts and ratios of different object types(utility, context, random).

---

**Example Question of Full$_{ideal}$ Dataset**

**Question ID**: 1,
**Utility**: heating(source),
**Question**: Which of the following objects would be best suited for the purpose of "heating(source)"?
**Options**:

  (A) **object name**: Toaster, **mass**: medium, **temperature**: RoomTemp, **material**: Metal,
      **already in use**: reversible-using, **condition**: dirty

  (B) **object name**: CoffeeMachine, **mass**: heavy, **temperature**: RoomTemp, **material**: Metal
      **already in use**: free, **condition**: clean

  (C) **object name**: CoffeeMachine, **mass**: heavy, **temperature**: RoomTemp, **material**: Metal
      **already in use**: irreversible-using, **condition**: clean

  (D) **object name**: ButterKnife, **mass**: light, **temperature**: RoomTemp, **material**: Metal,
      **already in use**: free, **condition**: clean

**Correct Answer**: B

---

### F.2.2 Full$_{moderate}$ Dataset

This dataset consisted of 30 variations of 15K QnA pairs with varying option counts and various ratios of object types (utility, context, random). Here, the correct answer would be the most appropriate `moderate configuration` of the `context object`. The other options could include

context objects(worse moderate and bad configurations), any configuration of objects compatible with the question's utility, and unrelated random objects (ideal, moderate, bad).

---

**Example Question of Full$_{moderate}$ Dataset**

**Question ID**: 1,
**Utility**: heating(source),
**Question**: Which of the following objects would be best suited for the purpose of "heating(source)"?
**Options**:

(A) **object name**: Toaster, **mass**: medium, **temperature**: RoomTemp, **material**: Metal, **already in use**: reversible-using, **condition**: dirty

(B) **object name**: CoffeeMachine, **mass**: heavy, **temperature**: RoomTemp, **material**: Metal **already in use**: reversible-using, **condition**: clean

(C) **object name**: CoffeeMachine, **mass**: heavy, **temperature**: RoomTemp, **material**: Metal **already in use**: irreversible-using, **condition**: clean

(D) **object name**: ButterKnife, **mass**: light, **temperature**: RoomTemp, **material**: Metal, **already in use**: free, **condition**: clean

**Correct Answer**: B

---

### F.2.3 Observations

| Dataset | PaLM Av. Accuracy | | | |
|---|---|---|---|---|
| | 2-opt | 3-opt | 4-opt | 5-opt |
| Ideal | 95.67 | 91.7 | 86.83 | 83.04 |
| Moderate | 77.83 | 62.08 | 55.03 | 48.68 |

Table 11: Average accuracy for **single prompt** evaluations in PaLM across variations of Full QA Dataset, i.e., accuracy averaged across various dataset variations for each fixed count dataset. The impressive performance of PaLM on F$_{ideal}$ dataset is no anomaly; we also saw its prowess in figuring out Ideal configurations of appropriate context objects even in Task-1 5. Further, we also observed in Task 2 how all language models (including PaLM) suffered when they were tasked to figure out suitable sub-optimal configurations. The same poor performance is witnessed once through PaLM's accuracy on F$_{moderate}$ dataset.

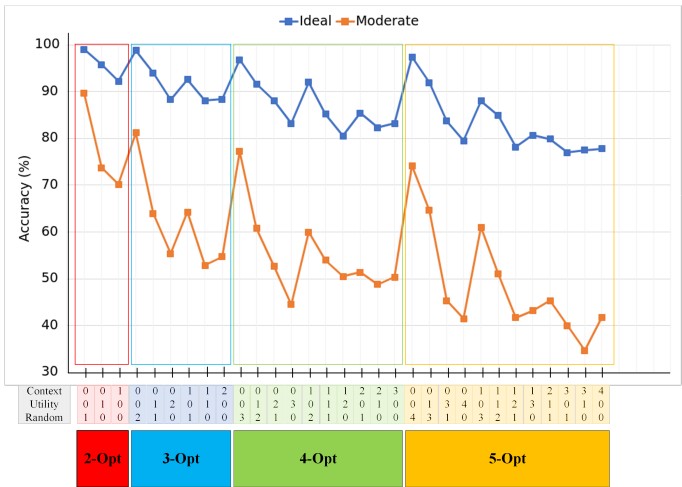

Figure 11: Variation Level analysis of PaLM model's accuracy across all variations of Full QA Dataset. For each dataset, 1 context object was set as the correct answer. Here (Context, Utility, Random) denote the count of each such object amongst the sampled options for each dataset variation.

Figure 11 plots the trend in accuracy as we vary the count of `<utility+random>` objects in the dataset options while increasing the `context` objects, thus increasing the level of difficulty. As expected (owing to the impressive utility level pruning capabilities), the peak accuracy for each fixed option count dataset occurs at maximum random objects. However, the worst accuracy was obtained when all the objects in the options were set as context options. We also observe improvement in accuracy whenever we increase the random object count or concept object count, supporting our previous conclusion of commendable object-utility mappings in language models like PaLM. In addition, PaLM's performance on the Ideal dataset is superior to its performance on the Moderate Dataset (just like we saw in task-1 and task-2 previously). Also, the observation of poor performance on $F_{moderate}$ dataset when all context objects are present aligned with our previous observations of our Task-2 ablations. (finding a suitable sub-optimal configuration amongst various sub-optimal configurations of context objects)6. Here, we could notice that each fixed count dataset is marked by a constant trend of accuracy drop whenever we move towards increasing the context objects - from left to right.

### F.3 Modular Setup

Owing to the below par performance of the PaLM language model on $F_{moderate}$ dataset, we experimented with a modular approach of breaking the question down into 2 levels as introduced in this work; `Object Level` and `Physical State Level`. This method consists of 2 parts:

1. **Object Selector**: We slice out the object names of the options and pass them as a separate question to the LLM. From here, we expect a list of objects(remember we could have multiple options consisting of configurations of context objects?) appropriate for the given `<Utility, Task>` combination.
2. **Selecting Physical States of Selected Objects**: On the basis of the object names received from stage-1, we slice out the options whose name belongs to that list and again call an LLM and ask it to analyze which option amongst those has a configuration that would be most suitable for the given `<Utility, Task>` combination. [8]

---

[8]To evaluate the merits of such technique, we test it out on $F_{moderate}$ dataset (as it had a wide margin for improvement, as shown in Figure 11 and Table 11:

| Prompt | PaLM Av. Accuracy $F_{\text{moderate}}$ | | | |
|--------|-------|-------|-------|-------|
|        | 2-opt | 3-opt | 4-opt | 5-opt |
| Single  | 77.83 | 62.08 | 55.03 | 48.68 |
| Modular | **81.96** | **70.11** | **61.52** | **54.37** |

Table 12: Average accuracy for **single prompt** and **modular** prompt evaluations in PaLM across variations of $F_{\text{moderate}}$ Dataset. Here, a single prompt means providing all option configurations in a single prompt to the language model for evaluation. We notice the increase in performance when we switch to a modular prompt regime across all fixed object count variations.

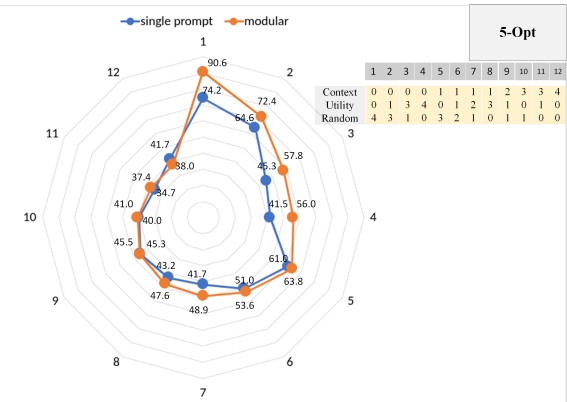

Figure 12: Comparative performance of **single prompt** method and **modular prompt** method implemented using PaLM and evaluated on 5-option variation of $F_{\text{moderate}}$ Dataset

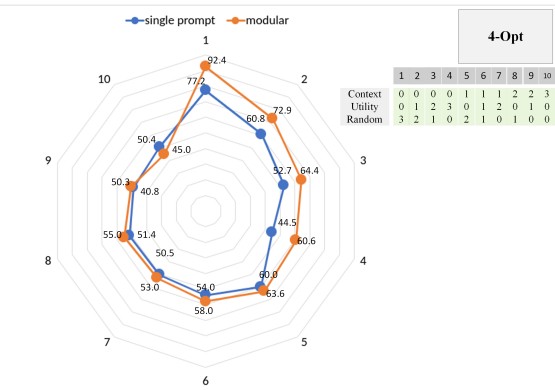

Figure 13: Comparative performance of **single prompt** method and **modular prompt** method implemented using PaLM and evaluated on 4-option variation of $F_{\text{moderate}}$ Dataset

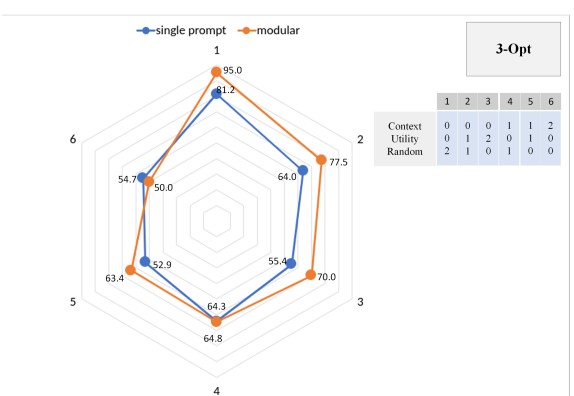

Figure 14: Comparative performance of **single prompt** method and **modular prompt** method implemented using PaLM and evaluated on 3-option variation of $F_{\text{moderate}}$ Dataset

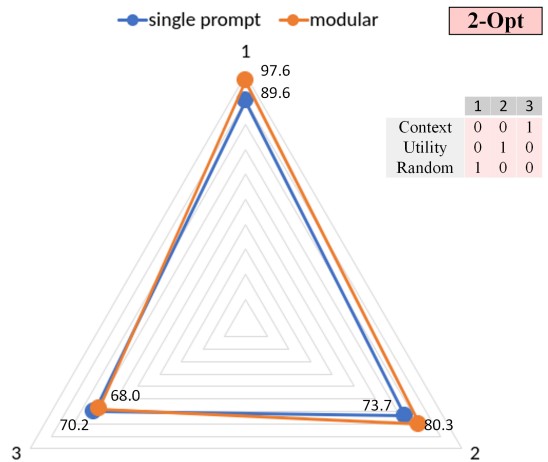

Figure 15: Comparative performance of **single prompt** method and **modular prompt** method implemented using PaLM and evaluated on 2-option variation of $F_{\text{moderate}}$ Dataset

### F.4 Observations

Figure 12,13,14 and 15 nightlight the improvement in performance across all variations except a few. The increase in the average accuracy for each type of fixed count dataset adds substance to our work's argument of breaking an object selection task into 2 broad phases (object selection and physical state selection). While the performance of PaLM was enhanced across nearly all variations, we also witnessed a few cases where there was a drop in accuracy(or no improvement). This was observed in datasets where only context objects were present in options, and thus, for such variations, modular categorization couldn't lead to performance gain. This is not a new observation; we had previously seen similar cases in single prompt technique (where evaluating such variations led to poor performance [See Orange plot in Figure 11]) and task-2 evaluations. However, an interesting thing observed when analyzing object selector LLM's responses was the confusion and random behavior it sometimes exhibited when tasked to output a list of correct objects in the presence of more than 1 correct object(context objects). For example, in the case of $F_{\texttt{ideal}}$ and $F_{\texttt{moderate}}$ experiments shown in Figure 11; or like even in task-1 or task-2 dataset questions - all these tasks could have options which contain multiple context objects corresponding to that `<Utility, Task>` combination. Thus, to tackle such cases, we must always prompt our object-level selector LLM to output us all the objects it considers appropriate for the given <Utility, Task> combination. Due to the random behavior of language models, we observed cases where one or more appropriate objects were discarded in the object-level stage. This often led to cases where the most appropriate sub-optimal configuration was discarded by rejecting the object name. (and thus its physical configuration couldn't be compared)

### F.5 Future Work

The next steps in this research direction would be fine-tuning the **physical state selector LLM** with task-1 and task-2 datasets to enhance its capabilities in judging object affordance given the physical state variables. In addition, fine-tuning the **object level selector LLM** with task-0 dataset and a multiple correct MCQ QnA (with outputs as lists of correct object names) would allow the object level responses to be more human behavior aligned. This would reduce the number of cases in the modular approach where the pipeline failed due to the failure of the object-level selector LLM. Future works would be aimed at comparing the modular approach consisting of such finetuned LLMs with **single prompting** method and **modular approach** employing off-the-shelf LLMs.

