# OpenReview forum: "Physical Reasoning and Object Planning for Household Embodied Agents"
_TMLR — Accepted by TMLR_

### Review · Reviewer_WAN4 · 2024-01-19

**Summary Of Contributions:**

This paper proposes an evaluation of language models according to how well they can perform question answering tasks of household object affordances. The paper introduces several datasets of Question Answering collected from human annotators. The datasets measure performance on three different tasks: “object-level pruning”, “variable level pruning task 1” (with ideal configuration present in the options), and “variable level pruning task 2” (with no ideal object configuration present in the options). Several LLMs are evaluated on variants of each task, which results in a set of performance comparisons between each. Broad trends of relative performance are visible in the results, as well as absolute performance trends that illustrate that current LLMs do not perform very well on these tasks, especially task 1 and 2.

**Audience:**

Yes

**Claims And Evidence:**

No

**Requested Changes:**

See the weaknesses for a list of proposed adjustments. Of those, the critical weaknesses are the absence of the evaluation of the full proposed pipeline, and the missing eval of one of the proposed phases.

**Strengths And Weaknesses:**

# Strengths
* To my knowledge, this is a new quantitative evaluation of LLMs that seems broadly relevant to deploying LLMs in household scenarios. Bluntly, if an LLM performed poorly on these tasks (as the current results show), it would likely lack a robust understanding of (human-centric) object affordances, and likely be unfit for any automated decision-making related to these tasks.
* The evaluation is multiple-choice question answering in every-day household settings, so it is reasonable to expect LLMs that claim to have broad general-purpose “knowledge” to exhibit reasonable performance on these tasks without fine-tuning (which the paper does not perform).
* I think the community would find the current middling performance of LLMs on these tasks interesting and motivating for future work.


# Weaknesses
* The evaluation appears to combine the “utility” and “context” phases. The paper needs justification either for why the phases are separated in Fig. 1, or why only the “context” phase is evaluated. E.g., “example task 0 variation 1” includes the “context” object “coffee”, whereas, based on my understanding of the “utility” phase, I’d expect an evaluation that does not include the context object (“coffee”) to be present somewhere, but it appears that it is not. If no context-free phase is evaluated, then I’d suggest changing the 3-phase modularization to 2-phase”.
* The evaluation appears to only test the modules for “context” and “physical state level”, instead of the entire 3-phase pipeline end-to-end (i.e., by using the outputs of each phase module as inputs to the next phase). This is a serious shortcoming that undermines the presence of the 3-phrase decomposition / modularization. Why does the paper introduce this modularization if it is not evaluated?
* Page 3: Unclear at this point in the paper what a “concept” is in “Object-Concept-Task mappings”
* Ideally, Fig. 1 should make the inputs to the ‘utility’ and ‘context’ stages clearer, partly to help justify separating these stages. I’d expect the inputs to the ‘context’ stage include some additional ‘context’ that is not available at the ‘utility’ stage, e.g., the first phase does not observe that the object to be cut is a “cake” but the second phase does observe this.
* The name “utility” is ambiguous because it’s also associated with the concept of “maximizing utility, since the intended output of the entire 3 phase decision-making process is the utility-maximizing object. Using the word “function” instead of “utility” would resolve this, as the first phase filters objects by their function, e.g., to objects that perform “cutting”. At the least, the text and figures that accompany the introduction of the name “utility” should clarify to readers this potential confusion.
* The utility annotation process is unclear (Appendix D and S 2.1.1). It’s not clear how  GPT3.5-turbo is used during annotation. D.1 states “Creation of utility-object mappings … involved the use of GPT3.5-Turbo”, yet only describes how the “annotators” (presumably all human) were involved. The paper needs to be more direct and specific about how GPT3.5-Turbo is used during annotation.
* S2.1.3, the acronym “LLM” is typically used to abbreviate “Large Language Model”, yet here it’s used to abbreviate “Language Learning Model”. This also conflicts with its usage on Page 2 (“large language models (LLMs)”)
* Fig. 5, 6 need y-axis labels.
* Fig. 7 has “Bad Accuracy” as the y-axis label, yet this was not introduced in the text. Is this the same as “Bad Rate”? The paper should clarify.
* The Fig. 7 caption should clarify that the dotted lines indicate the ratios of bad configurations to all configurations in each of the evaluations.
* The text refers to a Fig. 8, but there exists no Figure with a Figure 8 caption. I think the Fig. preceding S.4 needs a Figure number.
* The paper states “we generate human preference mappings and curate 2 QA consisting of 130K MCQ question-answer pairs” and the main contributions state “Introduction of three major novel CommonSense-based QA Datasets”. These statements seem inconsistent (2 != 3). The paper should clarify.
* The Dataset summary number of questions sums to 143.1K (= 68.9K + 58.7K + 15.5K), which is inconsistent with the statement  “we generate human preference mappings and curate 2 QA consisting of 130K MCQ question-answer pairs” (143.1K != 130K).
* The dataset is missing some description of the nature of the annotators, including how they were recruited and how they were incentivised to label.

---

> ### Author Response · Authors · 2024-02-20
> **Comments on Evaluation of Utility factor and Full Pipeline Evaluations**
>
> We thank reviewer WAN4 for their insightful comments and constructive feedback on our work. In addition to making the suggested edits in the manuscript (red text color), we have added experiments for full pipeline evaluation and added explanations for evaluating utility phase (blue text color Appendix F. Page 36 onwards). Below, we list the changes made:
>
> **Explanations regarding Utility Evaluations**: *We have added discussions on the 2 experiments we conducted before deciding to focus our experiments (task-0,1,2) entirely on Contextual factors and Physical State Factors.* [see page 36]
>  - **Qualitative** : We spawned an agent in AI2Thor and observed the RGB images annotated with the output of GPT3.5-Turbo (when prompted to give 2 utilities of every object that it comes across in the scene). Due to the high quality outputs, we concluded the presence of commendable Object-Utility associations in language models like GPT3.5-Turbo
>  - **Quantitative** : We also observed performance of PaLM and GPT3.5-Turbo on a 2K QA dataset designed to test object-utility alignment in LLMs. The near perfect results obtained from these experiments helped us conclude the presence of commendable Object-Utility associations in language models like PaLM and GPT-3.5-Turbo. Thus throughout the paper, we only focused on Contextual and Physical State Factors.
>
> **Discussion and Experiments related to Full Pipeline Evaluation and Modular Prompting**: *We curate 2 new datasets (F_ideal and F_moderate to evaluate the object selection capabilities in language models when tasked to employ all 3 factors (Utility, Context, Physical State) together.*[see Page 37 onwards]
>  - We further analyze the results obtained by evaluating PaLM language model on these 2 datasets. These results were observed to be in alignment with our previous conclusions of: a) Presence of *near perfect* object-utility mappings in models like GPT3.5-Turbo and PaLM [as observed in Utility Evaluations in Appendix F]  b.) Presence of *good* ideal configuration selection capabilities, *poor* moderate configuration selection capabilities, decreasing performance with increasing options. [as observed in task 1,2]
>
>  - We also discuss a **modular** prompting mechanism to divide the full question into 2 phases (object level and physical state level). We evaluate PaLM on F_moderate dataset and compare its performance when using **modular** vs **single** prompting technique. ("single" was default zero-shot method used throughout the work). We also analyze the increase in performance across individual datasets and discuss the shortcomings of current modular method. We also comment on ways to improvise such techniques.
>
> **Comments about nature of annotators and explanation of utility-object annotation process**:
>
> - Participant demographic spanned the university's various academic departments and consisted of students and researchers who volunteered for such annotations.
>
> - Creation of utility-object mappings that were further used as the backbone for all the tasks and datasets involved GPT3.5-Turbo and Human Annotation. This was done by using GPT3.5-Turbo to output Utilities for the 100 selected AI2Thor objects. From this, we then selected a random subset (after cross-checking it) and used it to create options while generating QnA to gather human annotation for Utility-Object Mappings. The annotators were asked to label 100 objects with utilities from a list of 22 utilities. These responses were then used for generating Utility-Objects mappings. These were further used as Ground Truth for the dataset used in "Utility Evaluations" in Appendix F.
>
> **Figure corrections and Typographical Edits** :
> - We have updated the figure to include the suggested edits.[Page 2]
> - Added a footnote informing the reader about the difference between the "utility" phase and the overall objective of "maximizing utility".
> - Improved the Figure 5 and 6 with y-axis
> - Added Figure heading in Figure 8 and explained dotted lines in caption.
> - Small typographical errors corrected and clarification regarding the count of datasets and their question count. [1 context level (around 15K) + 2 physical state level (around 130K) = 3 total datasets (around 145K)]
>
> Once again, we thank you for taking out time to review our work with such rigor and providing feedback to improve our paper -- we greatly appreciate it!!

---

> > ### Comment · Reviewer_WAN4 · 2024-03-15
> > **Concerns addressed, one new concern**
> >
> > The authors' response addressed my original concerns.
> >
> > However, the added utility evaluations on page 36 lack motivation in the main text, and the conclusion to not perform additional utility evaluations as a result is not discussed. The main text needs a clear logical connection between the result of the initial utility experiments and the motivation not to focus any remaining experiments on utility. It's not sufficient for the authors to state (as they did in their response) -- "Due to the high quality outputs, we concluded the presence of commendable Object-Utility associations in language models like GPT3.5-Turbo" -- the paper's main text needs to state something similar, and use it to motivate the focus of the experiments on the other factors. It's not sufficient to put the logical connection and motivation in the author response. The logical connection and motivation needs to be in the main text as well.

---

### Review · Reviewer_i1W5 · 2024-01-19

**Summary Of Contributions:**

This paper presents the CommonSense Object Affordance Task. This is a multiple-choice QA dataset that presents a task that needs an object with a certain utility to be completed, then asks what object or (object, physical state configuration) should be selected out of two to five options. The paper presents three considerations for such reasoning: the object's inherent utility, the contextual dependencies/appropriateness, and the current physical state, and designs its tasks accordingly. It also argues that humans use these considerations to make such judgments themselves. This is the primary motivation for the paper. The contributions are human annotations on COAT, the datasets themselves, and LLM baselines.

The paper goes through the creation of datasets, specifically how questions and answer options are generated. There are three tasks: one for eliminating objects based on contextual appropriateness, two for eliminating objects based on physical state configurations. Each dataset has thousands of questions. The Object-level dataset presents objects without the required utility, objects with the required utility but contextual inappropriateness, and objects with the required utility and contextual appropriateness. There are two physical configuration level datasets: one that has ideally-configured objects among the options, and one that has only moderately well-configured objects that would work but not ideally, and badly-configured objects that would not work. These present various challenges.

The experimental setup shows performance of 6-8 pretrained LLMs on answering MCQs from each of these datasets. The findings are that large models perform best, having more choices is generally harder, having a larger diversity of objects (not just physical configurations) is harder but bigger models are more robust to it, and having a higher percentage of bad configurations among the options leads to more frequent selection of bad configurations even when moderate configurations are always present and will be better.

The paper concludes with summarizing the results and suggesting that this data may be an avenue for improving the language model's abstract multi-step reasoning for estimating the physical affordance of everyday objects used in household activities.

**Audience:**

Yes

**Claims And Evidence:**

No

**Requested Changes:**

- Motivation: either find cognitive science literature that actually shows that humans use this type of reasoning (and ideally that it benefits them, but that's optional), or source motivation from somewhere else. In my opinion, it's okay to simply say this is a design that makes sense *if that design works* (which we don't know). Ultimately the paper needs results that show that training on COAT improves LLM performance, but even if there were something that could motivate this paper in the absence of these results, a claimed but unwarranted connection to human psychology doesn't.
- Tone down some of the other claims in the paper. For example, "extremely low-level utilities that could be assigned to a high level task" - the example given is "task: cutting cake", and "utility: cutting". I disagree that the task is much higher level than the utility in this case, and probably in a lot of cases (though there are certainly some where it is, such as helping a person with the shower task vs. cleaning utillity)
- Give more examples of the considerations, especially the contextual awareness one.
- Show examples of questions that LLMs actually succeeded/failed on, not just examples that illustrate the concept. It's not surprising that Bad and Ideal are separable, but it's interesting that Moderate is separable from them (and the Bad Rate result shows that it's not totally separable from Bad, so it would be nice to see examples to build our intuition of what, exactly, LLMs succeed and fail on)
- Add experiments training or fine-tuning LLMs on these tasks.

**Strengths And Weaknesses:**

#### Strengths
- Problem choice is important. Multi-step reasoning and physical reasoning are both things LLMs struggle with. The class of approach, a dataset and task, also makes sense - it has been clear for a long time, but seems to become increasingly clear, that the dataset is one of the most important factors in building large models.
- The dataset is well-designed. Given the motivating principles, it executes them very well in a way that's simple, extensible, and applicable to other domains. If training on this dataset led to benefits for LLMs, it would be a good point for expansion into more objects and settings, as well as a good template for building more datasets in difficult domains.
- The Task/Utility+Object-Utility-Configuration framework, plus the Ideal-Moderate-Bad configuration split, leads to rich and complex problems with opportunity to perform important ablations, from a compact set of factors.
- Physical state breakdown is intuitive
- Task construction is strong and well-explained. Like I said above, there are clearly lots of operations for testing different aspects of physical reasoning in a methodical way with this setup. The sub-optimal configuration dataset is especially interesting.
- Paper is overall very well-written clear. Paper is easy to understand (in a good way!) Only exception is the conclusion - it's one big paragraph of text right now, would be better to break it up. The writing itself is good.
- Figures could use more annotation but overall are well-presented
- Experimental section backs all of the claims made in the conclusion, and is good to see. It's helpful to know where LLMs are right now, and this experiment set is thorough. It's also clear that there is plenty of margin for improvement on such a task.

#### Weaknesses
**Motivation**
- The intro is predicated on the idea that humans reason in the exact way the paper lays out: first, they look for object utility (as defined and populated by this paper), then they look at contextual appropriateness (a wide, vague consideration, and also as defined by this paper), then they look at physical state. The boundaries between each of these three are not clear either. Overall, I find this breakdown very unintuitive and unnatural: for example, the clean knife case. What if there was a dirty knife? Where would that be eliminated - in the very broad "contextual appropriateness" phase? And the scissor indeed has a "cutting" utility, but is a scissor actually meaningfully capable of the type of cutting involved in cutting a cake, any more than a stick is? I realize that the real world is physically and semantically far more complex than any realistic set of systematically collected annotations can hope to capture, but 1) that doesn't mean gross simplifications work and 2) the entire intro is predicated on this being a human-like method of reasoning.
- The paragraph "to illustrate... current physical state" is extremely presumptuous about how humans reason. I genuinely cannot relate to this type of reasoning (as demonstrated in my scissor analysis above), and while I'm one person, there doesn't seem to be cognitive science literature backing any of these specifics - the only psych/cogsci references are related only at a high level and do not discuss such a reasoning framework in meaningful detail. This then makes it difficult to take the motivation seriously.
- I recognize that the Li et al. 2023 result talks about reasoning over physical variables being unhumanlike, and the paper asserts that the abstraction of the physical variables to its configuration framework is meant to make these considerations more humanlike, but we don't have proof that it *is* humanlike.

**Dataset creation**
- The contextual appropriateness consideration makes sense at a high level but is difficult to understand concretely. The paper says it "engaged human annotators in a study designed to assess the selection of appropriate objects for specified tasks and utilities" - this needs far more explanation. Examples, frameworks, what the humans were asked, etc. This is a very vague consideration that overlaps (in my mind, at least) with both utility and physical state, and it would help to understand it better.

**Experiments**
- Figures, though nice looking and readable, would benefit from being annotated with numbers (at least axes). The tables are there, but the axes labels should be there as well - that would be a better way to deliver the message.
- Ultimately, this paper does not show us that training on this dataset improves LLMs. We need that result for it to be meaningful, since there is no theoretical backing. The human motivation wouldn't really help even if it were warranted, because we are trying to improve LLMs' QA performance and not their efficacy as models of human psychology. That experiment needs to happen. Of course this is a big ask, but it is what it is. Right now, the paper is an idea - a fleshed out one with all the tools ready to go, but not a proven one (and not one that is well-motivated another way).

---

> ### Author Response · Authors · 2024-02-20
> **Response to Reviewer i1W5 (1/3)**
>
> We thank the reviewer i1W5, for their insightful comments and constructive feedback on our work. In addition to making the suggested edits in the manuscript (red text color), we have added modular prompting evaluations to highlight the performance gain when LLMs are prompted in a manner similar to our pipeline. We also have added the fine-tuning results.(blue text color Appendix F. Page 37 onwards) Below, we list the changes made:
>
> **Comments on Motivation of the work**
>
> We agree that there isn't much literature discussing the exact reasoning framework. However, our main motivation and sole focus of this work was to evaluate the object selection capabilities on the basis of **factors** that affect human choice during object selection (object level and physical state level). We have added experiments to highlight the benefit of such breakdown.(Page 40 onward) Regarding the example of "dirty scissors", we asked the LLM such questions during physical state level evaluations.
>
> ***
>
> **Clarifications on Figure-1** : We believe that Figure 1 might have caused some confusions and thus we have updated the figure. We explain it below:
>
> - Any object selection done by humans would consist of carefully considering factors such as utility,context and appropriate physical state; and thus to illustrate what we meant by each of such factors, we expressed it in the form of a pipeline as shown in Figure 1. (we show the benefit of such pipeline through the results of the **modular** prompting technique on Page 41)
> - We broke the decision making process into 2 major steps : **Object**  level analysis and **Physical State** Level Analysis.
>
>  **Object Level** : Comprised of 2 sub-levels of selection: "Utility" and "Context".
>
> Please note that **Utility Evaluations**(added in Appendix F) and **Context Evaluations**[Task 0]  _solely_ focused on evaluating Object Level Selection Capabilities (with no mention of the physical state of the object i.e., assuming every object is in its ideal state)
>  - **During the Utility level** : we provide “Utility” and object names as options (without any information about physical configuration) to judge their compatibility with “utility”(aspect of consideration) in question. Note that we also don’t give the information about the “task”. From here we get **"utility objects"**
>  - **During the Context level** : we provide “Task” in addition to “Utility” to select "context" objects. Here again, we give object names as options (without any information about their physical state). The options in each such question belongs to "utility objects" (for the task,utility asked in question). From here we get **"context objects"**.
>
> **Physical State Level**
>
> - During “Physical State” Level, we provide “task”, “utility” and “context objects” that we got from the last step (object-level step). Using these 3 information, we choose the most appropriate object configuration.
>
> - **Task-1** and **Task-2** focused  on Physical Level Selection Capabilities (of only the context objects i.e. precluding any errors arising due to wrong object selection)
>
> ***
>
> **Combined Reasoning**
>
> While Task-0 and Task 1,2 evaluated where either object level or physical state level was alone required, we have added experiments where both types of reasoning was required. [Page 37 onward]
> - 2 new datasets : **F_moderate** and **F_ideal** (newly added in Appendix F) focused on evaluating LLM capabilities in situations where application of both types of such reasoning was required. We refer to these experiments as **full pipeline evaluations**.
> - While we report the performance of PaLM on these datasets, we additionally discuss our observations (Page 40) and plot the accuracy trend across all variation of these 2 datasets (Figure 13).
>
> ***
>
> **Benefits of Pipeline**
>
> - While the full pipeline evaluations were done using a single prompt input, we also conducted experiments to evaluate the performance of PaLM when prompted in a manner similar to our **pipeline** (Figure 1). We refer to this as **modular** prompting.
> - We evaluated the **modular** prompting mechanism by dividing the full question into 2 phases (object level and physical state level, similar to how we designed our pipeline).
> - We report the findings when PaLM was evaluated on the F_moderate dataset and further compare these results to the **single** prompting method (asking the full question in one prompt)
> - Further, we analyze the increase in performance across individual datasets and discuss the current shortcomings and ways to improve upon these.  [see Table 12 and Figure 14,15,16,17]

---

> ### Author Response · Authors · 2024-02-20
> **Response to Reviewer i1W5 (2/3)**
>
> **Dataset Creation**
>
> Annotators with college-level education were recruited for this study. We broadly explain the type of questions asked during object-level and physical state-level:
>
> **During Object-level Analysis** :
>
> - Firstly human annotators were asked to mark all the "utilities" an object can be used for. (for object-utility mappings).
> - Secondly, they were asked to mark the objects that could be used for “utility” for a particular “task”("utility" to be focused on in the task).
> - For both steps we provided them with text-based options that consisted of utility names and object names. While first stage options were generated using GPT3.5-Turbo, we utilized the utility-objects obtained from 1st stage to generate options for 2nd stage.
>
> We illustrate it using an example:
>
> - **Utility Level**
>
>    Question: "StoveBurner" could be used for which of the following functions/utilities?
>
>    Options: [List of 22 utilities]
>
> - **Context Level**
>
>    Question: What object can be used as a “heating source” when tasked to make noodles?
>
>    Options: [A list of objects that were associated with the utility of "heating(source)"]
>
> **During Physical State level Analysis** :
>
> - Firstly, we obtained commonly occurring household configurations of all objects from the annotators. We gave an object name and asked them to choose all commonly occurring variable values of that object.
> - Secondly, we provided these commonly occurring physical states of the selected **context object** depending on the aspect of the task (i.e. task,utility combination). The annotators were then asked them to divide the configurations as "Ideal," "Moderate" and "Bad.". Example below:
>
>     Question: Classify  physical states of “StoveBurner” as Ideal, Moderate and Bad for the purpose of “heating(source)” in the task of making noodles.
>
>     Options: [A list of physical configurations of the object “StoveBurner” ]
>
> ***
>
> **Misc. Comments**
> -  *[Regarding Li et al. 2023]*: We wished to convey that our variables differ from them as ours are intuitive and natural language-based.
> - *[Regarding Figures]*: We've added y-axis labels to Figures 5 and 6 [Please see Pages 13, 14]
> - *[Toning down some claims]* : We've incorporated this change. [Please see Page 4 red text]
> - *[Improving Writing]*: We’ve improved the presentation of the conclusion section and split it into 3 paragraphs to make it easier to follow. [Page 15,16,17]
> -  *[Examples of Contextual Awareness]*: A task in addition to a utility adds context to the question. Below we give more clarity on contextual awareness:
>    - Objects associated with the **Heating Source** utility could be: Candle, StoveBurner, Microwave, Oven, etc. But we must mention a task to decide what objects would be usable. For example:
>        - We could use a candle for melting envelope wax, but we can’t use it to boil water
>    - Objects associated with **Cleaning** utiity could be: Towel, Sponge, Tissue Paper, etc. but we must mention a task to decide what objects would be usable. For example:
>      - We could use a _towel_ for cleaning our face, but can’t use a _Sponge_ for the same. Similarly, cleaning spilled ketchup using a _towel_ is inappropriate but can be done using _tissue-paper_.
>
> ***
>
> **Comments about fine-tuning**
>
>  - As requested, we performed the process of fine-tuning. We have added the exploratory results in the updated manuscript [Page 37] Our results highlight a promising increase of an average of 1.5-2% increase in object level selection performance and 7-10% increase in physical state level selection performance. As these performances were observed by fine-tuning on a very small fraction of our datasets and for a very few training steps, we expect these numbers to increase significantly when fine-tuned on a larger fraction of dataset for a larger number of training steps.

---

> ### Author Response · Authors · 2024-02-20
> **Response to Reviewer i1W5 (3/3)**
>
> **Regarding Observations & Intuitions**: We summarize our observed patterns below:
>
>   **Object level**
>    - **From Utility Evaluations** : We observed impressive abilities of language models to select objects on the basis of utility (see Appendix F. Page 36)
>    - **From Task-0 (Context Based)** : We observed decent abilities of language models to select objects on the basis of contextual factors. [Table 3]
>
>   **Physical State Level**
>    - **From Task-1** : Larger LLMs like PaLM and GPT3.5-Turbo showcased impressive abilities and outperformed smaller LLMs in identifying **Ideal** configurations. Smaller models performed moderately while choosing Ideal configurations (task-1). However, this was better as compared to their performance when tasked to choose most appropriate moderate configurations.(task-2)
>    - The dropping accuracy with the increase in object diversity further shed light on an existing bias towards commonly used objects. (this motivated LLMs to choose moderate configurations of more commonly used objects over Ideal configurations of equally appropriate but less commonly used objects. Example: moderate configuration of Watering Can was preferred over Ideal configuration of Bowl for carrying water while watering plants)
>    - **From Task-2** : We observed an increase in accuracy for all language models when we increased the count of bad configurations in options. [Table 5]
>   - However, **bad rates** differentiated between large and small LLMs [Table 6]. Large LLMs had very low bad rates even with increasing bad configurations. This led us to the conclusion that large LLMs like PaLM and GPT3.5-Turbo have commendable abilities to differentiate between moderate from bad configurations.
>   - However, the rise in bad rates for smaller language models highlights their inability to choose moderate configurations over bad configurations with increased fraction of bad configurations. (Poor performance in differentiating moderate from bad configurations as compared to larger language models).
>
> Once again, we thank you for taking the time to review our work with such rigor and providing such insightful comments to improve the study -- we greatly appreciate it!!

---

### Review · Reviewer_RSFx · 2024-02-06

**Summary Of Contributions:**

This paper introduces a new framework that aims to analyze common LLM's reasoning ability when comes to selecting objects for household tasks.The authors decide to format this decision process into three phases, each pruning out options based on utility, context and physical state. To quantitatively investigate LLM's performance on these household tasks, they come up with a Utility-object mapping by using GPT3.5 and human annotations. Then they create a dataset with QnA questions with progressive difficulty aiming to stress test LLM's reasoning ability.

**Audience:**

Yes

**Claims And Evidence:**

Yes

**Requested Changes:**

Overall this paper is well-motivated, and propose useful framework and datasets for future work to study the same question. It would be great if the authors could give more insightful analysis in the result section.

**Strengths And Weaknesses:**

Strengths
* This paper is well motivated. I believe the question they  study is a practical and important task for embodied agents.
* The proposed decision-making framework  seems logical and resembles human's decision process.
* The writing is clear and easy to follow.

Wealnesses
* I don't quite understand Figure 5. What's on the y axis? Similar to Figure 6.
* More analysis could be given in the result section. What other observations do the authors have besides the accuracy? Is there a pattern they see when different language models fail? Is there a reason that why PaLM performs much better than GPT3.5 with Task 1 and Task 2?
* If they give more in-depth analysis of the results, we might draw more useful conclusions or insights about how future work could improve the household tasks addressed in this paper.

---

> ### Author Response · Authors · 2024-02-20
> **Elaboration on Result Section**
>
> We thank reviewer RSFx for their insightful comments and constructive feedback on our work.
>
> **Regarding Figures**: We have updated Figure 5 and Figure 6 with y-axis labels for better understanding. These plots highlight the individual trend of all language models across different variations of datasets of task-1,2.
>
> **Regarding Observations & Intuitions**: We summarize our observed patterns below:
>
>   **Object level**
>    - **From Utility Evaluations** : We observed impressive abilities of language models to select objects on the basis of utility (see Appendix F. Page 36)
>    - **From Task-0 (Context Based)** : We observed decent abilities of language models to select objects on the basis of contextual factors. [Table 3]
>
>   **Physical State Level**
>    - **From Task-1** : Larger LLMs like PaLM and GPT3.5-Turbo showcased impressive abilities and outperformed smaller LLMs in identifying **Ideal** configurations. Smaller models performed moderately while choosing Ideal configurations (task-1). However, this was better as compared to their performance when tasked to choose most appropriate moderate configurations.(task-2)
>    - The dropping accuracy with the increase in object diversity further shed light on an existing bias towards commonly used objects. (this motivated LLMs to choose moderate configurations of more commonly used objects over Ideal configurations of equally appropriate but less commonly used objects. Example: moderate configuration of Watering Can was preferred over Ideal configuration of Bowl for carrying water while watering plants)
>    - **From Task-2** : We observed an increase in accuracy for all language models when we increased the count of bad configurations in options. [Table 5]
>   - However, **bad rates** differentiated between large and small LLMs [Table 6]. Large LLMs had very low bad rates even with increasing bad configurations. This led us to the conclusion that large LLMs like PaLM and GPT3.5-Turbo have commendable abilities to differentiate between moderate from bad configurations.
>   - However, the rise in bad rates for smaller language models highlights their inability to choose moderate configurations over bad configurations with increased fraction of bad configurations. (Poor performance in differentiating moderate from bad configurations as compared to larger language models).
>
> Once again, we thank you for taking the time to review our work with such rigor and providing such insightful comments to improve the study -- we greatly appreciate it!!

---

### Decision · Action_Editor_UcMT · 2024-04-09

**Recommendation:** Accept with minor revision

**Comment:**

There were mixed opinions among the reviewers, but having taken a closer look, the AE is satisfied that the work holds value, even if the manuscript lacks some polish, and leaves open some obvious room for improvement, such as training on these datasets to improve LLM reasoning abilities (even though within this work, the datasets are mainly curated for evaluating pre-trained LLMs rather than for improving them).

The AE has suggested minor revisions above: namely, improve the introduction to tone down claims about human decision making that are not established in the appropriate literature, and incorporate a brief description of the findings from the newly added appendix to motivate the experiment design. Besides this, a thorough grammar and typo check is also recommended.

**Audience:**

Yes, the AE thinks this is a valuable addition to the ongoing community efforts to understand what LLMs know (or at least what knowledge can be easily elicited from them) and what they do not.

**Claims And Evidence:**

The paper's claims are *mostly* supported by the evidence it presents.

The paper is best understood as an empirical evaluation of where LLMs stand in terms of reasoning about the right objects to use for various household tasks. The setup to study this involves curating three question-answering datasets and the key findings are that LLMs perform only moderately well at this task.

The experiment design is good enough that the findings will be of value to researchers. However, claims about human decision making that inspired the experiments indeed appear poorly supported, as i1W5 argues. This is in my view somewhat tangential to the key findings of the paper and easily fixed in a minor revision.

WAN4 notes that the main paper does not in fact evaluate the object-utility associations directly, only the "context" and "state". In response, the authors have added experiments in Appendix F.1. However, as WAN4 notes, these should at least be referenced within the main text: without at least a summary of this evidence, the experiment design is incoherent with the introduction.